

# Upwelling and isolation in oxygen-depleted anticyclonic modewater eddies and implications for nitrate cycling

Johannes Karstensen[1], Florian Schütte[1], Alice Pietri[2], Gerd Krahmann[1], Björn Fiedler[1], Damian Grundle[1], Helena Hauss[1], Arne Körtzinger[1,3], Carolin R. Löscher[1], Pierre Testor[2], Nuno Viera[4], Martin Visbeck[1,3]

[1]GEOMAR, Helmholtz Zentrum für Ozeanforschung Kiel, Düsternbrooker Weg 20, 24105 Kiel, Germany
[2]LOCEAN, UMPC; Paris, France
[3]Kiel University, Kiel Germany
[4]Instituto Nacional de Desenvolvimento das Pescas (INDP), Cova de Inglesa, Mindelo, São Vicente, Cabo Verde

*Correspondence to*: Johannes Karstensen (jkarstensen@geomar.de)

**Abstract.** The physical (temperature, salinity, velocity) and biogeochemical (oxygen, nitrate) structure of an oxygen depleted coherent, baroclinic, anticyclonic mode-water eddy (ACME) is investigated using high-resolution autonomous glider and ship data. A distinct core with a diameter of about 70 km is found in the eddy, extending from about 60 to 200 m depth and. The core is occupied by fresh and cold water with low oxygen and high nitrate concentrations, and bordered by local maxima in buoyancy frequency. Velocity and property gradient sections show vertical layering at the flanks and underneath the eddy characteristic for vertical propagation (to several hundred-meters depth) of near inertial internal waves (NIW) and confirmed by direct current measurements. A narrow region exists at the outer edge of the eddy where NIW can propagate downward. NIW phase speed and mean flow are of similar magnitude and critical layer formation is expected to occur. An asymmetry in the NIW pattern is seen that possible relates to the large-scale Ekman transport interacting with ACME dynamics. NIW/mean flow induced mixing occurs close to the euphotic zone/mixed layer and upward nutrient flux is expected and supported by the observations. Combing high resolution nitrate ($NO_3^-$) data with the apparent oxygen utilization (AOU) reveals AOU:$NO_3^-$ ratios of 16 which are much higher than in the surrounding waters (8.1). A maximum $NO_3^-$ deficit of 4 to 6 µmol kg$^{-1}$ is estimated for the low oxygen core. Denitrification would be a possible explanation. This study provides evidence that the recycling of $NO_3^-$, extracted from the eddy core and replenished into the core via the particle export, may quantitatively be more important. In this case, the particulate phase is of keys importance in decoupling the nitrogen from the oxygen cycling.

## Introduction

Eddies are associated with a wide spectrum of dynamical processes from the mesoscale (order of several 10 to 100 km) to the submesoscale (order of 10 meters to less than 1 km). The interaction of these processes creates transport patterns in and around eddies that provoke intense biogeochemical and biological feedback (Levy et al. 2012, Chelton et al. 2011a). At the ocean surface, the eddy rotation generates a sea level anomaly (SLA) pattern that allow for their remote detection (Chelton et al. 2011a).



But also in other surface parameters, such as seas surface temperature (SST) or chlorophyll-a fluoresence, eddies show anomalies that allow deriving their statistics (Chelton et al. 2011b; Gaube et al. 2015). Utilizing satellite data large/global scale analysis of eddy-generated anomalies has been conducted (Chaigneau et al. 2009, Chelton et al. 2011a, Gaube et al. 2015). More recently the vertical

structure of mesoscale eddies have been studied on regional (Chaigneau et al. 2011) and global scale (Zhang et al. 2013) combining eddy surface detection with concurrent, but opportunistic, in-situ profile data (e.g. Argo floats). These studies differentiate, based on the SLA being either positive or negative, cyclonically rotating and anti-cyclonically rotating eddies.

However, when vertical stratification is considered, a third group of mesoscale eddies emerges that

combine the downward displacement of isopycnals towards the eddy centre, as observed in "normal" anticyclonic eddies (AE), with the upward displacement of isopycnals, characterizing cyclonic eddies (CE). Such "hybrid eddies" are called anticyclonic mode-water eddies (ACME) or intra-thermocline eddies (McWilliams 1985, D'Asaro 1988, Kostianoy and Belkin 1989, Thomas 2008), because the depth interval between upward and downward displaced isopycnals creates a volume of rather

homogenous properties, sometimes called "mode" water. A recent study (Schütte et al. 2015) investigated the occurrence of these three different types of mesoscale eddies (CE, AE, and ACME) in the eastern tropical North Atlantic, combining in-situ profile data with satellite SLA and SST data. The authors found that ACME in the tropical eastern North Atlantic are characterized by a cold SST anomaly (in contrast to AEs that show a warm SST anomaly), which allowed a statistical assessment of

ACME in the tropical eastern North Atlantic based on satellite data alonse. Schütte et al. (2015) estimated that about 9% of all eddies (20% of all anticyclones) in the eastern tropical North Atlantic are ACME.

More than a decade ago a dedicated observational program was carried out to survey eddies in western North Atlantic (Sargasso Sea) in order to better understand the physical-biogeochemical interactions.

The surveys revealed that ACME showed particularly intense deep chlorophyll-a layers that aligned with a maximum concentrations of diatoms and maximum productivity (McGillicuddy et al. 2007). The high productivity was linked to a conceptual model that explains the intense vertical fluxes of nutrients within ACME with an Ekman divergence generated from the horizontal gradient in wind stress across an ACME (McGuillicuddy et al. 2007; going back to a work from Dewar and Flierl, 1987). The concept

itself was questioned (Mahadevan et al. 2008) and high-resolution ocean model simulations, comparing runs with or without eddy-wind interaction, reproduced only a marginal impact on ocean productivity (but a strong impact on physics; Eden and Dietze 2009). However, from a tracer release experiment within an ACME the magnitude of the vertical flux was found to be of the magnitude as expected from simple eddy-wind induced Ekman pumping estimates (order of meters per day; Ledwell et al. 2008).





A measure for the importance of local rotational effect relative to the Earth rotation is given by the Rossby number defined as $Ro = \frac{\zeta}{f}$, and where $\zeta = \frac{\partial V}{\partial x} - \frac{\partial U}{\partial y}$ is the vertical vorticity ($U, V$ are zonal and meridional velocity) and $f$ the planetary vorticity. Planetary flows have small $Ro$ say up to ~0.2, while local rotational effects gain importance with $Ro$ approaching 1. Mesoscale eddies often have $Ro$ close to

1 indicating that, for example, the geostrophic approximation does not hold locally. Anticyclonic rotating eddies (AE and ACME) have negative relative vorticity ($\zeta < 0$) and modify $f$ into an effective planetary vorticity ($f_{eff} = f + \frac{\zeta}{2}$) (Kunze 1985, Lee and Niller 1998). This local reduction in $f$ has for example implications for the propagation of internal waves that occur at frequencies in the range of the local buoyancy frequency (N) and $f$. For AEs, the local reduction of $f$ leads to a trapping of internal

waves of near inertial frequencies (NIW) and downward energy propagation inside of AEs (Kunze 1985, Gregg et al. 1986, Lee and Niller 1998, Koszalka et al. 2010, Joyce et al. 2013, Alford et al. 2016). Depending on the vertical distribution of N the downward NIW propagation can generate an amplification of the wave energy and eventually part of that energy is dissipated by critical layer absorption which in turn results in intense vertical mixing (Kunze 1985, Kunze et al. 1995, Whitt and

Thomas 2013).

The anticyclonic rotation of ACME also reduces $f$ but, in contrast to AE, a local N maximum at their upper boundary characterizes the ACME. This N maximum creates a "cap" and complicates NIW vertical propagation. Lee and Niller (1998) considered an ACME stratification in a model study on near inertial internal wave propagation, mimicking the stratification of a "California subsurface warm core

eddy", and found the accumulation of energy below the eddy. A recent study (Sheen et al. 2015) reported on hydrography, dissipation measurements, and current observations across an ACME observed at 2000 m depth in the Antarctic Circumpolar Current. The study found enhanced diapycnal mixing at the periphery of this particular ACME. Applying a ray tracing method to a stability profile from outside and from inside of the eddy Sheen et al. showed that, depending on their propagation

direction relative to the background shear, the modelled internal waves either encounter a critical layer above and below the eddy or are reflected at the eddy core.

A number of studies analysed NIW interaction with Mediterranean Outflow lenses in the North Atlantic ("Meddies", Armi and Zenk 1984), which are deep ACME (mode below 1000m). Krahmann et al. (2008) reported observations of enhanced NIW energy at the rim of a Meddie. For Meddies signatures

of layering at the eddy periphery have often been observed and related to the interaction of NIW with the baroclinic shear of the rotating lens, ultimately driving dissipation (Hua et al. 2013). The dissipation is often associated with Kelvin-Helmholtz instabilities (e.g. Kunze et al. 1995) but other mixing processes, such as double-diffusion may play a role too. In particular layering, produced by internal wave shear and strain, has been also reported to enhance double diffusive fluxes (St Laurent and



Schmitt 1999), at least in regions that are already characterized by salt finger-favourable stratification (Kunze 1995).

Levy et al. (2012) summarized the submesoscale upwelling at fronts in general, not specifically for mesoscale eddies, and the impact on oceanic productivity. Vertical property flux of nutrients into the
euphotic zone plays a key role in the productivity scenarios and can be either by advection or by mixing. Regardless of the mechanism for vertical transport, mesoscale eddies have been found to be "retention regions" (d'Ovidio et al. 2013) and the upwelling of properties into the mixed layer, either inside or at the periphery of mesoscale eddies, is trapped in the eddy where it can be utilized efficiently.

Recently, ACME with very low oxygen concentrations in their cores where observed in the tropical
eastern North Atlantic (Karstensen et al. 2015). The generation of the low oxygen concentrations was linked to high productivity in the euphotic zone of the eddy and the subsequent remineralisation of the sinking organic matter, a process that requires a vertical fluxes of nutrients to supply the productivity in the euphotic zone. However, the eddy core were found to be highly isolated against surrounding waters and weak transport and mixing activity in and around the core was expected. As such it is surprising
that the isolated core can exist in parallel to the enhanced vertical flux, and that suggested that the processes responsible for the separation must act on small spatial scales. In this paper we further investigate the transport processes associated with a low oxygen ACME making use of high-resolution underwater glider and ship data. This paper is part of a series of publications that report on different genomic, biological and biogeochemical aspects of low oxygen ACME in the eastern tropical North
Atlantic (Löscher et al. 2015, Hauss et al. 2016, Fiedler et al. 2016, Schütte et al. 2016).

## 2 Data and Methods

Targeted eddy surveys are logistically challenging. Eddy locations can be identified using real-time satellite SLA data. To further differentiate a positive SLA (indicative for anticyclonic rotation) into either an AE or an ACME the SST anomaly across the eddy was inspected, because ACME in the
eastern tropical North Atlantic show a cold SST anomaly (Schütte et al. 2015, Schütte et al. 2016). For further evidence, but purely opportunistic, Argo profile data were inspected to detect anomalously low temperature/salinity signatures, indicative of low oxygen ACME in the region (Karstensen et al. 2015, Schütte et al 2016). In late December 2013 a candidate eddy was identified through this mechanisms and in late January 2014 a pre-survey was initiated, making use of autonomous gliders. After
confirmation that the candidate eddy was indeed a low oxygen ACME, two ship surveys (ISL, M105; Fiedler et al. 2016) and further glider surveys followed.





## 2.1 Glider survey

Data from the glider IFM13 (1st deployment) and Glider IFM12 (2nd deployment) were used in this study (Fig. 1). IFM12 surveyed temperature, salinity, and oxygen to a depth of 500 m while IFM13 surveyed the same properties down to 700 m. Both glider measured chlorophyll-a fluorescence and

turbidity to 200 m depth. In addition to the standard sensors IFM13 was also equipped with a nitrate sensor that sampled to 700 m depth. IFM13 surveyed one full eddy section from April 3-7, 2014 (Fig. 1). For IFM12 we combined data from February 3-5 and from February 7-10, 2014 because, due to technical problems, data were not recorded in between these periods. All glider data was internally recorded as a time series along the flight path, while for the analysis the data was interpolated onto a

regular pressure grid of 1 dbar resolution. For the purposes of this study we consider the originally slanted profiles as vertical profiles.

## 2.2 Glider Sensor Calibrations

Both gliders were equipped with a pumped CTD and no evidence for further time lag correction of the conductivity sensor was found. Oxygen was recorded with AADI Aanderaa optodes (model 3830). The

optodes were calibrated in reference to SeaBird SBE43 sensors mounted on a regular ship-based CTD, which in turn were calibrated using Winkler titration of water samples (see Hahn et al. 2014). The calibration process also removes a large part of the effects of the slow optode response time via a reverse exponential filter (time constants were 21 and 23 seconds for IFM12 and IFM13, respectively). As there remained some spurious difference between down and up profiles we averaged up- and

downcast profiles to further minimize the slow sensor response problem in high gradient regions, particular the mixed layer base.

The nitrate measurements on IFM13 were collected using a Satlantic Deep SUNA sensor. The SUNA emits light pulses and measures spectra in UV range of the electromagnetic spectrum. It derives the nitrate concentration from the concentration-dependent absorption over a 1 cm long path through the

sampled seawater. During the descents of the glider the sensor was programmed to collect bursts of 5 measurements every 20 seconds or about every 4 m in the upper 200 m and every 100 seconds or about every 20 m below 200 m. The sensor had been factory calibrated 8 months prior to the deployment. The spectral measurements of the SUNA were post-processed using Satlantic's SUNACom software, which implements a temperature and salinity dependent correction to the absorption (Sakamoto et al., 2009).

The SUNA sensors' light source is subject to aging which results in an offset nitrate concentration (Johnson et al., 2013).

To determine the resulting offset, nitrate concentrations measured on bottle samples by the standard wet-chemical method were compared against the SUNA-based concentrations. The glider recorded profiles close to the CVOO mooring observatory (see Fiedler et al. 2016) at the beginning and end of



the mission. These we compared to the mean concentrations of ship samples taken in the vicinity of the CVOO location (Fiedler et al. 2016). In addition we compared glider measurements within the ACME to nitrate samples from two surveys performed during the eddy experiment (see Löscher et al. 2015, Fiedler et al. 2016). The comparison showed on average no offset ($0.0 \pm 0.2$ µmol kg$^{-1}$). However, near

the surface the bottle measurements indicated nitrate concentrations below 0.2 µmol kg$^{-1}$ at CVOO, while the SUNA delivered values of about 1.8 µmol kg$^{-1}$ possibly related to technical problems near the surface. We thus estimate the accuracy of the measurements to be better than 2.5 µmol kg$^{-1}$ with a precision of each value of about 0.5 µmol kg$^{-1}$.

All temperature and salinity data is reported in reference to TEOS-10 (IOC et al. 2010) and as such we

report absolute salinity ($S_A$) and conservative temperature ($\Theta$). Calculations of relevant properties (e.g. buoyancy frequency, spiciness, oxygen saturation) were done using the TEOS-10 MATLAB toolbox (McDougall and Barker, 2011). We came across one problem related to the TEOS-10 thermodynamic framework when applied to nonlinear, coherent vortices. Because the vortices transfer properties nearly unaltered over large distances the application of a regional (observing location) correction for the

determination of the absolute salinity (McDougall et al. 2012) seems questionable. In the case of the surveyed eddy the impact was tested by applying the ion composition correction from 17°W (eddy origin) and compared that with the correction at the observational position, more than 700 km to the west, and found a salinity difference of little less than 0.001 g kg$^{-1}$.

### 2.3 Ship survey

In between the two glider surveys, a ship survey of the respective eddy was conducted on the 18th and 19th March 2014 (Fig. 1) with R/V METEOR (expedition M105), about 6 weeks after IFM12 and 3 weeks before IFM13. We make use of the water currents data recorded with a vessel mounted 75kHz Teledyne RDI ADCP device. The data was recorded in 8 m depth cells and standard processing routines where applied to remove the ship speed and correcting the transducer alignment in the ship's hull. The

final data was averaged in 15 min intervals. Only data recorded during steaming (defined as ship speed larger than 6 kn) is used for evaluating the currents structure of the eddy. It should be mentioned that the inner core of the eddy shows a gap in velocity records, which is caused by very low backscatter particle distribution (size about 1 to 2 cm) (see Hauss et al. 2016 for a more detailed analysis of the backscatter signal including net zooplankton catches).

In order to provide a frame to compare ship currents and glider section data, we interpolated data from 8 deep (>600 m) CTD stations performed during the eddy survey, and estimated oxygen and density distributions across the eddy section. More information about other data acquired during M105 in the eddy is given elsewhere (Löscher et al. 2015, Hauss et al. 2016, Fiedler et al. 2016).



### 3 Results and Discussion

#### 3.1 Vertical Eddy Structure

In order to compare the vertical structure of the eddy from the three different surveys, all sections were referenced to "kilometre distance from the eddy core" as the spatial coordinate, while the "centre" was

selected based on visual inspection. Comparing the oxygen concentrations from the three surveys reveals a core of low oxygen in the centre of the eddy (Fig. 2). The core extends over a depth range from about 60 to 200 m depth in all three surveys and its upper and lower boundaries are aligned with the curvature of isopycnals. Considering the whole section across the eddy it can be seen that towards the centre of the eddy the isopycnals show an upward bending in an upper layer (typically associated

with cyclonic rotating eddies) and a downward bending below (associated with anticyclonic rotating eddies) which is characteristic for ACME.

During the first survey (IFM12), lowest oxygen concentrations of about 8 µmol kg$^{-1}$ were observed in two vertically separated cores at about 80 m and 120 m depth, while in between the cores oxygen concentrations increased to about 15 µmol kg$^{-1}$. About 6 weeks later, during METEOR M105, lowest

concentrations of about 5 µmol kg$^{-1}$ were observed, centred at about 100 m depth and without a clear double minimum anymore. During the last survey (IFM13), another three weeks later, the minimum concentrations were < 3 µmol kg$^{-1}$ and now showed a single minimum at 120 m. It is unknown how much of the intensification of the minimum (by about 5 µmol kg$^{-1}$ in 2 months) is attributed to sampling the core during the individual surveys at different distances from the core. The broadening and

deepening however, seems to be a real signal. Underneath the eddy core, and best seen in the 40 µmol kg$^{-1}$ oxygen contour below 350 m, a slight increase in oxygen over time is found probably related to lateral exchange and the general increase in oxygen towards the west (ventilated gyre region).

The horizontal and vertical extent of the low oxygen core in the two glider surveys is rather similar (Fig. 2), perhaps a little larger during the second survey. The gridded oxygen contour from the ship

survey, based on eight stations, suggest a narrower core. However, these differences in the core characteristic may also be related to the survey having been made a few kilometres further away from (or closer to) the eddy centre. The similarity in the vertical structure suggests that the general eddy structure and the observed phenomena are stable across the three surveys, as expected for a coherent eddy (McWilliams 1985). A composite of the outermost ("last") closed geostrophic contour of the eddy

(Fig. 1, right), analysed from a selected set of SLA data (see Schütte et al. 2016 for details), revealed a diameter of about 60 km, which is in accordance with the dimensions of the vertical structures observed from the glider and the ship (Fig. 2 and 3).

Defining the low oxygen core by oxygen concentrations below the canonical value of 40 µmol kg$^{-1}$, we find that the core is composed of a fresh (and cold; not shown) water mass (Fig. 3a) that matches the



characteristics of South Atlantic Central Water (SACW; Fiedler et al. 2016), which is typical for the low oxygen eddies in the eastern tropical North Atlantic (Karstensen et al. 2015, Schütte et al. 2015, 2016). These properties indicate that the ACME was formed in the coastal area off Mauritania (see Fiedler et al. 2016, Fig. 1, left) and that the eddy core did not experience significant exchange with the

surrounding waters. The core water mass characteristic is well expressed in the spiciness (McDougall and Krzysik 2015) section that shows the contrasting impact of $\Theta$ and $S_A$ on isopycnals (Fig. 3b). The core is well separated in spiciness from the surrounding waters, while the core itself shows only weak vertical structure in spiciness and a nearly horizontal orientation of isopycnals.

The low oxygen core of the ACME is well separated from the surrounding water through a maximum in

the squared buoyancy frequency ($N^2$; Fig. 3c) and as such in stability. The most stable conditions are found along the upper boundary of the core ($N^2 > 10 \cdot 10^{-5}$ s$^{-2}$) aligned with the mixed layer base. Here, vertical gradients in $\Theta$ ($S_A$) of 5 K (0.7 gr kg$^{-1}$) over vertical distances of 15 m are observed. At the lower side of the ACME the stability maximum is less strong ($N^2$ about $3 \cdot 10^{-5}$ s$^{-2}$) but separating the eddy surrounding water well from the low $N^2$ of the core (the "mode"). The stability also shows, as seen

in $S_A$, slanted vertical bands of alternating stability patterns at the rim and below the eddy.

The pattern are also evident in the stability ratio $R_\rho = \frac{\alpha^\theta \Theta_z}{\beta^\theta (S_A)_z}$, (here shown as a Turner angle; Tu; McDougall and Barker 2011) (Fig. 3d). The $R_\rho$ is the ratio of the vertical contribution from $\Theta$, weighted by the thermal expansion coefficient ($\alpha^\Theta$), over that from $S_A$, weighted by the haline contraction coefficient ($\beta^\Theta$), to the static stability (Fig. 3d). For convenience $R_\rho$ is converted to Tu

using the four-quadrant arctangent. For Tu between -45 to -90° the stratification is susceptibility to salt finger type double diffusion, while Tu 45° to 90° indicate susceptibility for diffusive convection. Regions were most likely double diffusion occurs are found for Tu close to ±90°. In the core of the eddy (Fig. 3d) the Tu indicate that diffusive convection is possible (Tu values close to +45°), however, no gradients exists and no fluxes are expected here. In contrast, below the core in the alternating and

vertically slanted Tu patterns, values within ±45° to ±90° can be seen. The structures do not align with the tilting of the isopycnals but cross isopycnals. Moreover, salinity (and temperature) gradients are seen in these depth ranges as well (see Fig. 3a) and salt finger fluxes is possible. A Tu between −75° to −85° at 120 m, 160 m, 210 m depth (salt finger susceptible) and Tu between 80° and 89° at 210 m, 320 m, 410 m depth (diffusive convection susceptible).

The ADCP zonal (Fig. 4a) and meridional (Fig. 4b) currents show a baroclinic, anticyclonic rotating flow, with a maximum swirl velocity of about 0.45 m s$^{-1}$ at about 100 m depth. The maximum rotation speed (approximately represented by the zonal section) decreases nearly linear to about 380 m depth where 0.1 m s$^{-1}$ is reached. Alternating currents with about 80 to 100 m wavelength can be seen close to the eddy edges and more clearly seen after subtracting 120 m boxcar filtered profiles (Fig. 4 c). The



local (at 19°N) inertial period is 36.7 h while the ADCP section was surveyed in 14 h (including station time) and only a moderate aliasing effects is expected in the sections. In contrast, the glider took more than 5 days (4 inertial periods) to complete the section (Fig. 3) and a mixture of time/space variability is mapped.

Considering the translation speed of the eddy of 3 to 5 km day$^{-1}$ (see Fiedler et al. 2016) the nonlinearity parameter $\alpha$, relating maximum swirl velocity to the translation speed, is much larger than 1 (about 6.5 to 11 in the depth level of the low oxygen core) and indicates a high coherence of the eddy. At the depth of the maximum swirl velocity, and considering the eddy radius of 30 km, a full rotation would take about 5 days but for the deeper levels more.

## 3.2 Eddy core isolation and vertical fluxes

The concept proposed for the formation of the low oxygen zone is based on an isolation of the eddy core, in combination with high productivity and subsequent respiration of sinking organic material – in analogy to the formation of dead-zones in coastal and limnic systems (Karstensen et al. 2015). The water mass observed in the ACME defines a strong anomaly against the surrounding waters and is very

similar to water masses at the eastern boundary, where the ACME was formed (Fiedler et al. 2016). The constancy of hydrographic properties over a period of about 7 month have been shown for this ACME (Fiedler et al. 2016) and suggests that no significant exchange between eddy core and surrounding water through either lateral or diapycnal processes occurred.

The high nonlinearity parameter for the eddy ($\alpha \gg 1$) indicates its coherence and explains why the

eddy was so remarkably stable when comparing observations of the eddy 2 month apart (Fig. 2). However, the high $\alpha$ does not explain the isolation of the core against mixing. Mixing in the thermocline is closely related to breaking of internal waves (Gregg et al. 1986, Gregg 1989), maybe with local enhancement by double diffusive mixing in cases where vertical gradients are intense (St Laurent and Schmitt 1999). To determine the possible interaction of the low oxygen eddy with the

internal wave field we first calculate the relative vorticity $\zeta$ of the eddy (Fig. 5a) and then investigated the impact on the local $f$. The anticyclonic (negative) $\zeta$ in the core of the low oxygen eddy is about $-0.8 \cdot f$ ($Ro = -0.8$) and changes sign at the boundary of the core, where large $Ro$ remain ($Ro \sim 0.4$) suggesting that eddy rotational effect are important. The $f_{eff}$ in the ACME (Fig. 5b) is lowered to values of 0.6 in the core (Kunze et al 1995), while underneath the core, and below the lower N$^2$

maximum, still values of 0.8 are found. Such low $f_{eff}$ force NIW to propagate downward also called "inertial chimney" (Lee and Niller 1998). In contrast to a typical AE stratification (Kunze 1985, Kunze et al. 1995, Joyce et al. 2013) the low oxygen ACME has an intense N$^2$ maximum ($10^{-4}$ s$^{-2}$) defining its upper boundary and that also aligns with the mixed layer base (Fig. 3c). Model studies (Lee and Niiler 1998) and ray tracing analysis (Sheen et al. 2015) show for ACME that most of the NIW energy



accumulates below the eddy. As a consequence, mixing inside the core is low and is well separated from mixing outside the core. This separation of mixing regimes we interpret as why the low oxygen eddy core has been so constant in its water mass properties.

Having now the isolation explained as a consequence of eddy rotation and stratification and their joint impact on the propagation of internal waves, it is tempting to investigate whether the vertical flux of nutrients into the euphotic layer of the eddy may also fit to this concept. The productivity associated with the ACME requires intense episodic or prolonged moderate upward fluxes of nutrients into the euphotic zone. From global assessments of productivity in mesoscale eddies based on satellite data (Chelton et al. 2011b; Gaube et al. 2015) the upwelling is identified in the centre of the eddies. Note, that these studies do not explicitly consider ACME. Schütte et al. (2016) showed that low oxygen ACME do have productivity maxima (indicated by enhanced ocean color based Chlorophyll-a estimated) at the rim of the eddies, not at the centre. This may suggest that the vertical flux is also concentrated to the rim. Inspecting the glider section data (Fig. 3) does not show indications for upwelling in the centre, isopycnals in the core are flat and the core has only a weak (and vertical) stratification. The mixed layer base is characterized by the very stable stratification and large gradients (e.g. 0.3 K m$^{-1}$ in temperature). The ship's thermosalinograph temperature data (Fig. 3a upper) show the eddy to be colder than the surroundings but also indicate that a local maximum in temperature is observed in the centre of the eddy, while local minima (0.2 K difference) at ±15 km distance from the centre. Considering the first glider oxygen section (IFM12, Fig. 2a), the upper of the two separate minima is found very close to the depth of the mixed layer base and indicate that any exchange across the mixed layer by mixing processes must be very small.

Following up on the NIW discussion, we now consider the region outside the core and in the vicinity of the eddy. It is a reasonable assumption that a large fraction of the NIW energy originates from wind stress fluctuations (D'Asaro 1985). The NIW propagation in the upper layer of the eddy (above the core) is primarily along the intense N contours/mixed layer base (Fig. 3c) and towards the eddy rim (shown in Sheen et al. 2015). At the eddy rim, but outside of the maximum N contour, the NIW can propagate downward (see Fig. 4 b,d) because $f_{eff} < 1$ (Fig. 5b). In this zone we observed the impact of the vertical propagation in the vertical shear profile directly via the modulation of the maximum swirl velocity (approximately Fig. 3a, at ~ −32 km distance).

In order to extract the velocity signal related to the NIW ($U'$), we subtracted the filtered (120 m boxcar filter) velocity profile data ($U_0$) from the observed profiles ($U$; Fig. 5c):

$$U = U_0 + U'$$

Magnitude and angle between the zonal and meridional NIW components of $U'$ show a vertically stacked pattern within the $f_{eff}$ region, suggesting the trapping of the NIW underneath the eddy (Lee



and Niiler 1998). The phase angle and its cyclonic rotation with depth indicate a downward propagation of the waves (Joyce et al. 2013). Another region where rotation and a maximum in amplitude is seen is the outer rim of the eddy in the region where $f_{eff}$ approaches 1, in the depth range between 50 and 120 m (~ −30 km). The NIWs have an amplitude of more than 0.1 m s$^{-1}$ and a vertical scale of about

70 to 90 m (Fig. 4d), similar to observations at mid latitude fronts (Kunze and Sanford 1984). The inertial radius for a wave with amplitude of 0. 1 m s$^{-1}$ is about 2 km. The NIW wave pattern underneath the eddy core aligns well with the pattern observed in the glider sections (Fig 2 and 3).

A Richardson gradient number $Ri_g = N^2/((\frac{\partial u}{\partial z})^2 + (\frac{\partial v}{\partial z})^2)$ cannot directly be estimated along the sections because no concurrent velocity and stratification section data exists. Considering single profile

data the shear in velocity is about 0.1 m s$^{-1}$ over 50 to 70 m depth (Fig. 3d) equals a variance of $0.2 - 0.4 \cdot 10^{-5}$ s$^{-2}$ that is implied along the wave propagation path. Outside of the mixed layer base N$^2$ maximum, a corresponding $Ri_g < 10$ can be expected to occur (see Fig. 2c), a value that may indicate generation of instabilities (Joyce et al. 2013).

Only vertical propagation of internal waves does not generate mixing, but the waves have to either

break (Kelvin-Helmholtz instabilities) or produce enough shear to generate critical layer absorption. When considering the maximum velocities (0.1m s$^{-1}$) associated with the NIW they account for about 25% of maximum swirl velocity. However, a region close to, but outside of, the maximum swirl velocity (about ~ −32 km, 50 to 120m depth) is identified where $f_{eff} < 1$ and the NIW velocity ($U'$) is of similar magnitude as the flow ($U$) and thus susceptible for critical layer formation. Here the mean

flow could gain energy from the NIW but also vertical mixing. It has been shown that trapping of NIW inside an AE (Gulf Stream warm core ring, Joyce et al. 2013) generated most instabilities and mixing close to surface and where most horizontal shear in the baroclinic current is found. In case of the ACME discussed here, this is close to the mixed layer base and at 90 m depth (Fig. 4a) and potentially opening a pathway for properties from below into the mixed layer (and vice versa). Some evidence for

such a flux can be seen in the nitrate section (Fig. 6 b) that show a local maximum maybe associated with an upward filament at about 100 m depth/distance of about −32 km.

An asymmetry in the dynamical structure is seen for example in the $f_{eff}$ section (Fig. 5b). All sections run from south (most negative distance) to north and it is the southern part that is more coherent and energetic and this is also the flank were the NIW signal is strongest. It is possible that the interaction of

the eddy with the north-easterly trade winds play a role here because the eddy frontal flow is in the direction of the wind. Thomas (2005) showed the generation of a vertical flux at the mixed layer base that entrains water from underneath.

Below the eddy the S$_A$ gradients (Fig. 3a) do align well with the wave crests and that may indicate the impact of intense strain, and an thus a periodic intensification of S$_A$ gradients which in turn could





enhance the susceptibility to double diffusive mixing in regions where susceptibility of double diffusion is already indicated by the Tu distribution (Fig. 3d).

### 3.3 A nutrient budget for the eddy

In order to interpret the low oxygen concentrations in terms of biogeochemical processes, we calculated the apparent oxygen utilization (AOU, Fig. 6a), which is defined as the difference between measured oxygen concentration and the oxygen concentration of a water parcel of the given $\theta$ and $S_A$ that is in equilibrium with air (Garcia and Gordon, 1992; 1993). AOU is an approximation for the total oxygen removal since a water parcel left the surface ocean. The low oxygen concentrations in the core of the eddy are equivalent to an AOU of about 240 µmol kg$^{-1}$ (Fig 6a). Along with high AOU we also find very high $NO_3^-$ concentrations with a maximum of about 30 µmol kg$^{-1}$ (Fig 6b). The corresponding AOU:$NO_3^-$ ratio outside the core is 8.1 and thus close to the classical 8.625 Redfield ratio (138/16; Redfield et al. 1938). However, in the core an AOU: $NO_3^-$ ratio of >16 is found. This high ratio indicates that less $NO_3^-$ is released during respiration (AOU increase) than expected for a process following a Refieldian stoichiometry. By considering a linear fit to outside the core (Fig. 6c) the respective $NO_3^-$ deficit can be estimated to up to 4-6 µmol kg$^{-1}$ for the highest AOU ($NO_3^-$) observations. By integrating $NO_3^-$ and $NO_3^-$-deficit over the core of the low oxygen eddy (defined here as the volume occupied by water with oxygen concentrations < 40 µmol kg$^{-1}$) we obtain a average AOU: $NO_3^-$ ratio of about 20:1.

One way to interpret this deficit is by $NO_3^-$ loss through denitrification processes. Loescher et al. (2015) and Grundle et al. (in revision) both found evidence for the onset of denitrification in the core of the ACME discussed here. Oxygen concentrations in the core are very low (about 3 µmol kg$^{-1}$) and denitrification is possible. Evidence for denitrification in the core of the ACME was, however, demonstrated as being important for $N_2O$ cycling at the nanomolar range (Grundle et al. in revision), and not necessarily for overall $NO_3^-$ losses which are measured in the micromolar range. Estimates of N* from the ACME show that even in the core of the ACME $NO_3^-$ losses were not detectable at the micromolar range (Fig. 6d). Thus, while denitrification may have played a minor role in causing the higher than expected AOU:$NO_3^-$ ratio which we have calculated, it is unlikely that it contributed largely to the loss of 5% of all $NO_3^-$ from the eddy as estimated based on the observed AOU:$NO_3^-$ ratios.

Alternatively, but perhaps not exclusively, the $NO_3^-$ recycling within the ACME could be the reason for the $NO_3^-$ deficit. A high AOU: $NO_3^-$ ratio could be explained through a decoupling of $NO_3^-$ and oxygen recycling pathways in the eddy. In this scenario $NO_3^-$ molecules are used more than one time for the remineralization/respiration process and therefore the AOU increase without a balanced $NO_3^-$ remineralization. Such a decoupling can be conceptualized as follows (Fig. 7): consider an upward flux of dissolved $NO_3^-$ and oxygen in a given ratio with an amount of water that originates from the low





oxygen core. The upward flux partitions when reaching the mixed layer, one part disperses in the open waters outside of the eddy, the other part is keep in the eddy by retention (D'Ovidio et al. 2013). The upwelled $NO_3^-$ is utilized by autotrophs for primary production and thereby incorporated into particles (PON) while the corresponding oxygen production is ventilated to the atmosphere. The PON sinks out

of the mixed layer/euphotic zone and into the core of the eddy were remineralization of organic matter releases quickly some of the same $NO_3^-$ back into the core. In contrast, the upwelling of oxygen-deficient waters will drive an oxygen flux from the atmosphere into the ocean in order to reach chemical equilibrium. But because the stoichiometric equivalent of oxygen is lost to the atmosphere and therefore not transported back into the core by gravitational settling of particles, as is the case for nitrate

(via PON), the respiration associated with the remineralization of the recycled nitrate will results in a net increase in AOU. A potential problem with this concept is that the $NO_3^-$/oxygen from the eddy core is primarily outward, so it must be more an erosion rather then a flux, because a flux would also transport from outside into the core and thus slowly altering the core properties (which is not observed). It could be an erosion process that maintains an stability maximum along with the exchange and maybe

related to interleaving (Beal 2007). Further work is needed to understand this process in full.

**4 Summary and Conclusion**

Here we present a first analysis of high-resolution multidisciplinary glider and ship survey data of a low oxygen anticyclonic mode-water eddy (ACME) in the eastern tropical North Atlantic. The eddy has a diameter of about 70 to 80 km and maximum swirl velocity of 0.4 m s$^{-1}$ (at about 90 m depth) and can

be considered typical for the region (Schütte et al. 2015, 2015; Karstensen et al. 2015). The eddy originated from the Mauritanian upwelling region (Schütte et al. 2016; Fiedler et al. 2016) and had a distinct, anomalously fresh (and cold), water mass in its core that was located immediately below the mixed layer base (about 70 to 80 m) and a depth of 200 to 250 m. The core showed minimum oxygen concentrations of 8 µmol kg$^{-1}$ during the first glider survey (February 2014) and 3 µmol kg$^{-1}$ during the

second glider survey, 9 weeks later. Enhanced productivity was estimated for the eddy (Fiedler et al. 2016), implying a vertical flux of nutrient rich waters to the euphotic zone/mixed layer. A concept for the isolation of the core but enhanced vertical flux of nutrients in parallel was derived (Fig. 7). The velocity observations indicate that the eddy had a very distinct impact on the propagation of near inertial internal waves as expected (Kunze 1985, Kunze et al. 1995, Sheen et al. 2015). The

combination of a negative vorticity anomaly of an anticyclonic rotating eddy and a maximum in stratification that encloses the ACME's low oxygen core on the one side traps the NIW to the eddy vicinity but also prevent NIW to enter the core. A velocity shear variance maximum is found below the eddy that is interpreted as an NIW energy maximum, as seen from model simulations of NIW propagation around ACME (Lee and Niiler 1998). Outside the low oxygen core, at the upper bound of

the stability maximum, we expect enhanced mixing to occur, as shown for ACME in the deep ocean



(Sheen et al. 2015). Moreover, for AE it has been reported that in the depth range were the NIW interact with the maximum baroclinic flow (in our case at about 90 m) enhanced mixing can occur (Joyce et al. 2013) possibly by critical layer formation. Our analysis suffers from not having concurrent hydrography and currents data, and limited options for estimating balances (e.g. Richardson numbers) exists.

However, if we transfer the findings by Joyce et al. (2013) to our eddy the enhanced mixing would be at the mixed layer base, but outside the eddy. An asymmetry in the dynamical structure of the eddy is observed (not so much in hydrography though) with a more intense front in the southern part of the section and which is directly under the impact of the Ekman flow generated by the Norteast Trade winds. A NIW induced mixing would create an upward flux of nutrients, also supported by the $NO_3^-$

distribution. Once $NO_3^-$ is in the mixed layer the eddy retention (D'Ovidio et al. 2013) will trap a fraction of the upwelling waters. The $AOU:NO_3^-$ ratio of the eddy core is altered high (16) when compared with the classical Redfield ratio (8.625) or the background ratio (8.1). We estimated the $NO_3^-$ deficit for the eddy which is about 1:20 when referenced to the total $NO_3^-$ content. Denitrification is one possible process but the significant nitrate loss of the core seems unrealistic. What is more likely is a

local recycling of N but not oxygen and connected to the transfer of upwelled $NO_3^-$ from the core via sinking of PON. The isolation of the eddy core in combination with high productivity is a prerequisite for the formation of the low oxygen core and as such analogue to the formation of a "dead-zones", known to occur in coastal and limnic systems (Karstensen et al. 2015). The isolated core is the rare case of an isolated volume of water in the open ocean and which allow to study fundamental biogeochemical

cycling processes in the absence of significant physical transport processes. A number of surprising biogeochemical cycling processes and ecosystem responses have been reported from the studies on eastern tropical North Atlantic low-oxygen eddies (Löscher et al. 2015, Hauss et al. 2016, Fiedler et al. 2016; Fischer et al. 2016, Grundle et al. in revision, Schütte et al. 2016). The NIW concept for the vertical flux outside the core but likewise the isolation of the ACME core that we presented here is

based on internal wave processes that are not routinely resolved by numerical models. A strategy for parameterizing of these processes is however required considering the estimate by Schütte et al. (2016) who showed that the enhanced respiration in low oxygen eddies contribute about 20% to net respiration that creates the shallow oxygen minimum of the eastern tropical Atlantic.

**Acknowledgment**

We thank the authorities of Cape Verde for the permission to work in their territorial waters. We acknowledge the support of the captains and crews of R/V Islandia (glider survey support) and R/V Meteor. We thank Tim Fischer (GEOMAR) for processing the ADCP data and Marcus Dengler for fruitful discussions. Financial support for this study was provided by a grant from the Cluster of Excellence "The Future Ocean" to J. Karstensen, A. Körtzinger, C.R. Löscher, and H. Hauss. Glider

data analysis where supported by the DFG Collaborative Research Centre754 (www.sfb754.de). B.





Fiedler was funded by the Germany Ministry for Education and Research (BMBF) project SOPRAN (grant no. 03F0662A). F. Schütte and P. Testor were supported by the trilateral project AWA supported by BMBF (grant no. 01DG12073E). Analysis was supported by European Union's Horizon 2020 research and innovation programme under grant agreement No 633211 (AtlantOS).

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




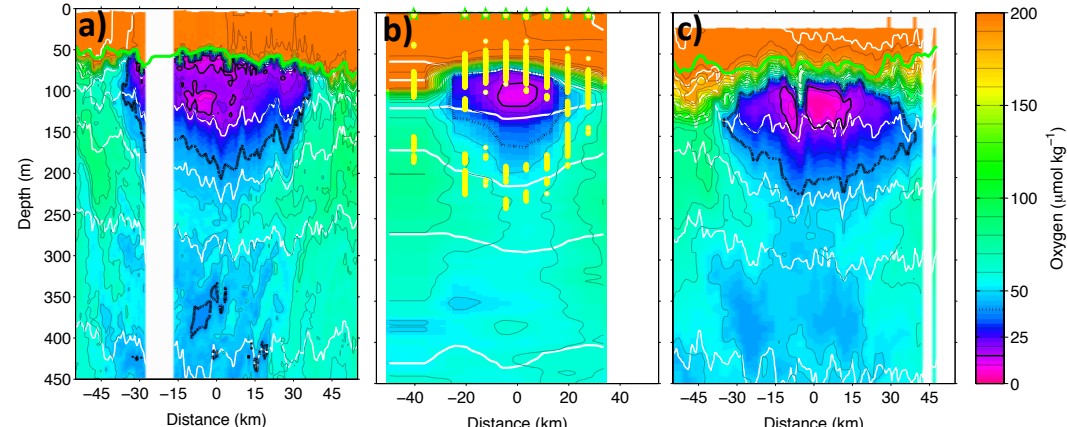

**Figure 2: Oxygen distribution from the three surveys (locations of the surveys see Fig. 1) over a similar distance (-55 to 55 km relative to a subjectively selected eddy centre) a) IFM12, b) M105, and c) IFM13. The 15 μmol kg$^{-1}$ (40 μmol kg$^{-1}$) oxygen contour is indicated as bold (stippled) line, selected density anomaly contours are shown as white lines (Δσ = 0.2 kg m$^{-3}$). The green line indicates the mixed layer depth. The oxygen contour in b) was gridded based on the 8 stations (locations indicated by green stars) and mapped to a linear section in latitude, longitude. The yellow dots indicate positions of local stability maxima.**




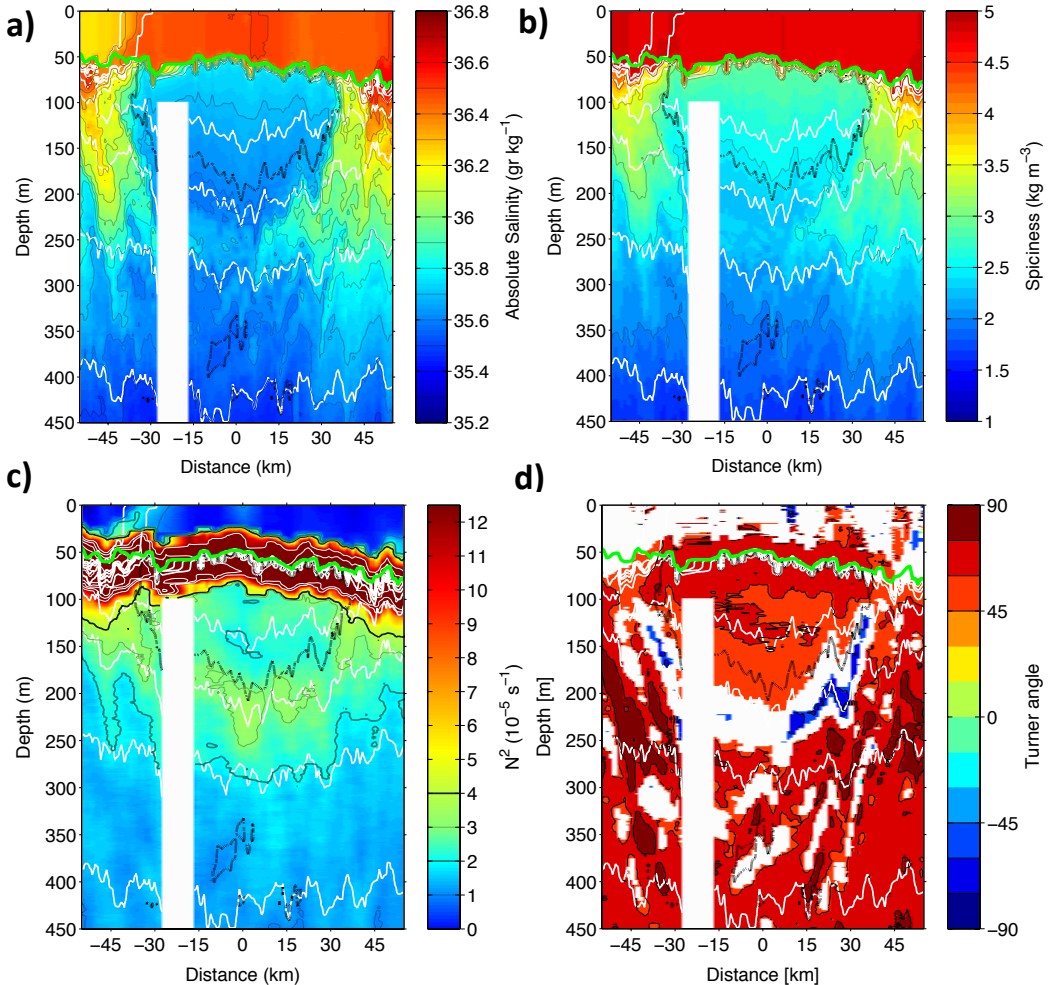

**Figure 3:** a) $S_A$, b) spiciness, c) buoyancy frequency/stability ($N^2$), d) Turner angle (only segment |45| to |90| is shown). All section are from IFM12. The thick black broken line indicates the 40 µmol kg$^{-1}$ oxygen concentration (see Fig. 2). Colour coding and selected contours see individual colour bar. Selected density anomaly contours are shown as white lines ($\Delta\sigma = 0.2$ kg m$^{-3}$). The green line indicates the mixed layer depth.




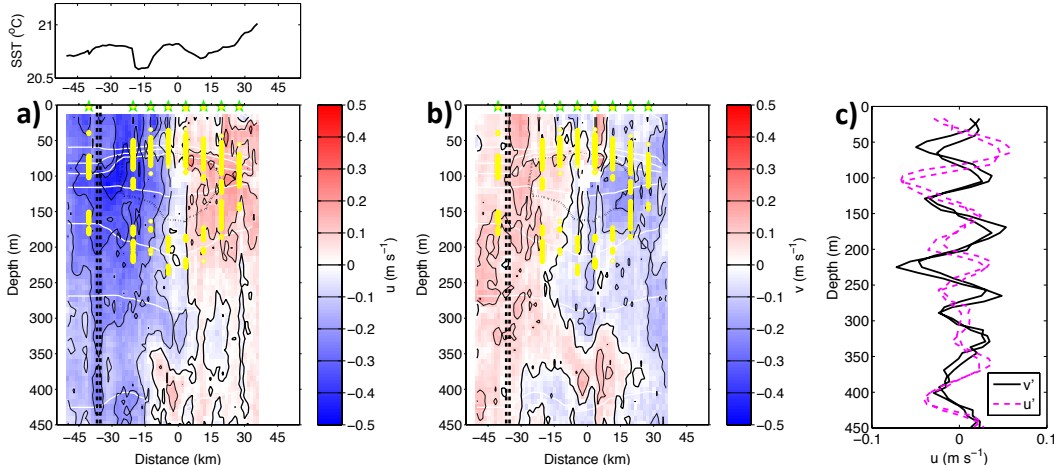

**Figure 4: a) Zonal velocity section (with thermosalinograph temperature on top), b) Meridional velocity section, c) Profiles (24 m**
5  **box car filtered) of residual velocity after subtracting a 120m box car profile from observed (8 m bin) ADCP profile data (for
location see vertical broken lines in a & b). In a & b the 40 µmol kg$^{-1}$ oxygen concentration (see Fig. 2) as well as selected density
anomaly contours (a, c, Δσ=0.2 kg m$^{-3}$) are shown, derived from gridded CTD profile data (station marked by green stars). In a, b
the yellow dots indicate the $N^2$ maximum from the CTD profiles.**





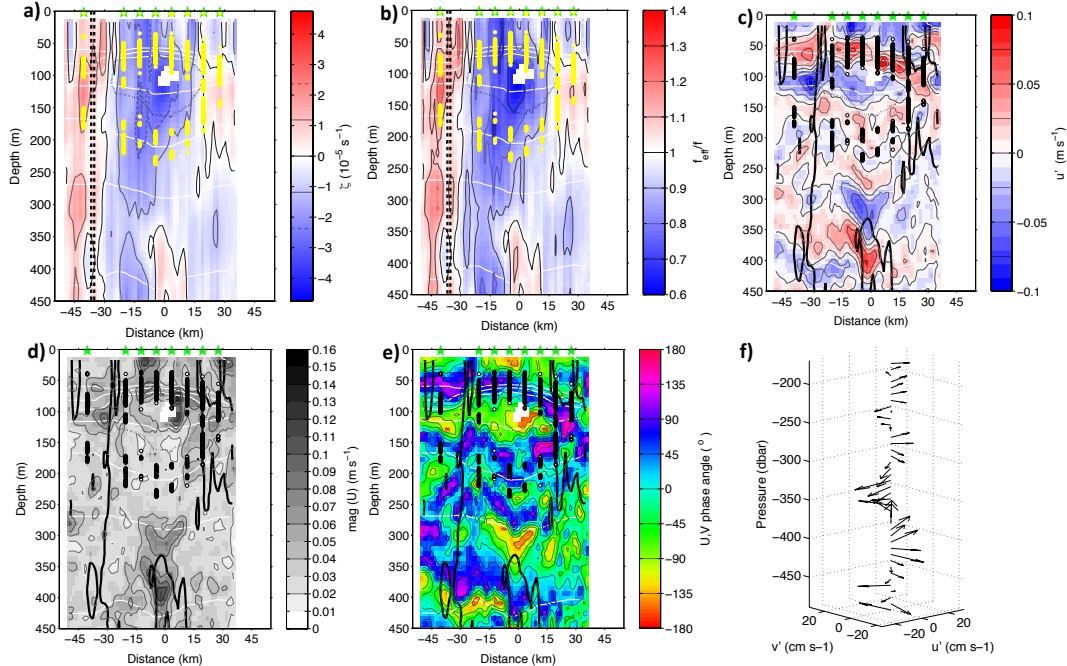

**Figure 5: From M105 survey: a) Vertical vorticity (colourscale covers the range +- $f$), b) effective planetary vorticity $f_{eff}$, c) zonal U' component (see also Fig. 4c for example profiles) d) magnitude of U' e) phase angle between u' and v' components, f) Phase angle at 0 km distance and only underneath the low oxygen core of the ACME. Vertical broken line in a), b) at about -32km distance indicate position of profiles Fig. 4c. For colour and line coding see individual colour bar. The yellow (black) dots in a, b (c, d, e) indicate the $N^2$ maximum from the CTD profiles. The white contours indicate selected isopycnals, black contour in c), d), e) is the $f_{eff}$=1 contour (See Fig. 5b).**




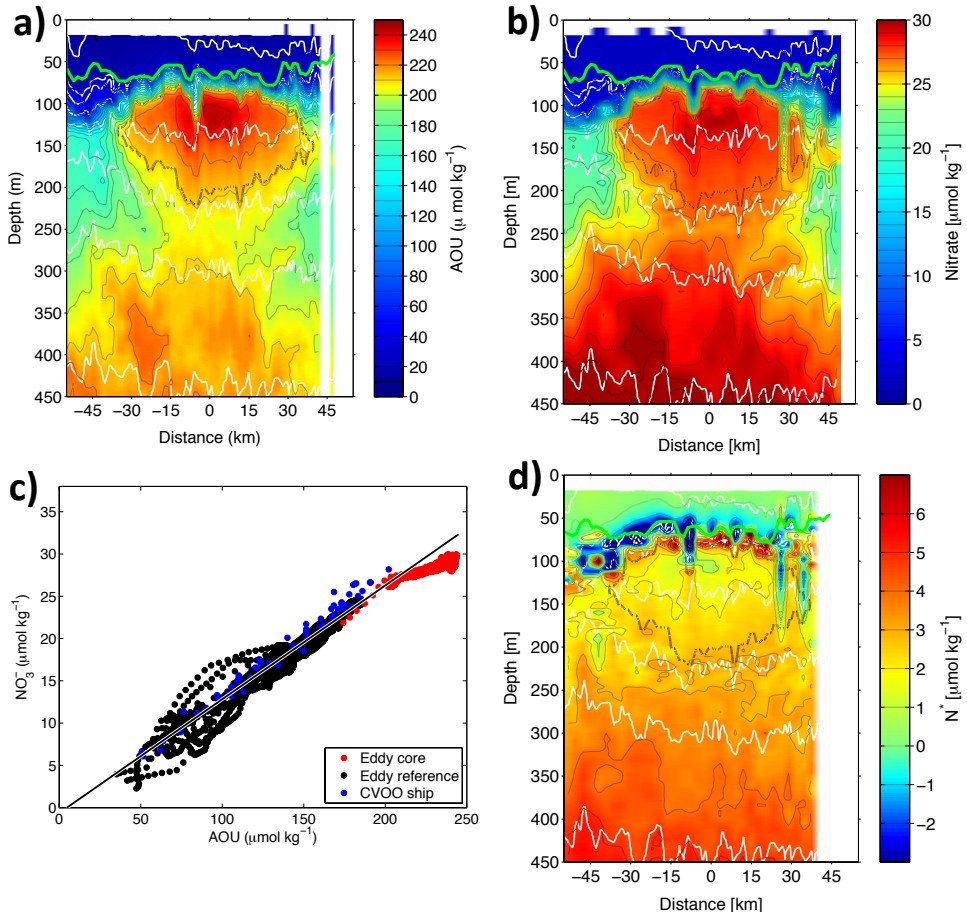

**Figure 6: a) AOU and b) NO$_3^-$ from IFM13 glider survey. c) AOU versus NO$_3^-$ for the depth range 90 to 250m depth: (red dots) IFM13 glider survey in the low oxygen core, (black dots) IFM13 glider survey close to CVOO, (blue dots) CVOO ship data (see Fielder et al. 2016). The black line is the linear fit to all data (slope 8.1) and used to estimate an NO$_3^-$ deficit in the low oxygen core.**

5 **d) N* estimated from NO$_3^-$, AOU/(8.625), note, negative values close to mixed layer are related to different sensor response time of oxygen optode and SUNA and not further discussed.**




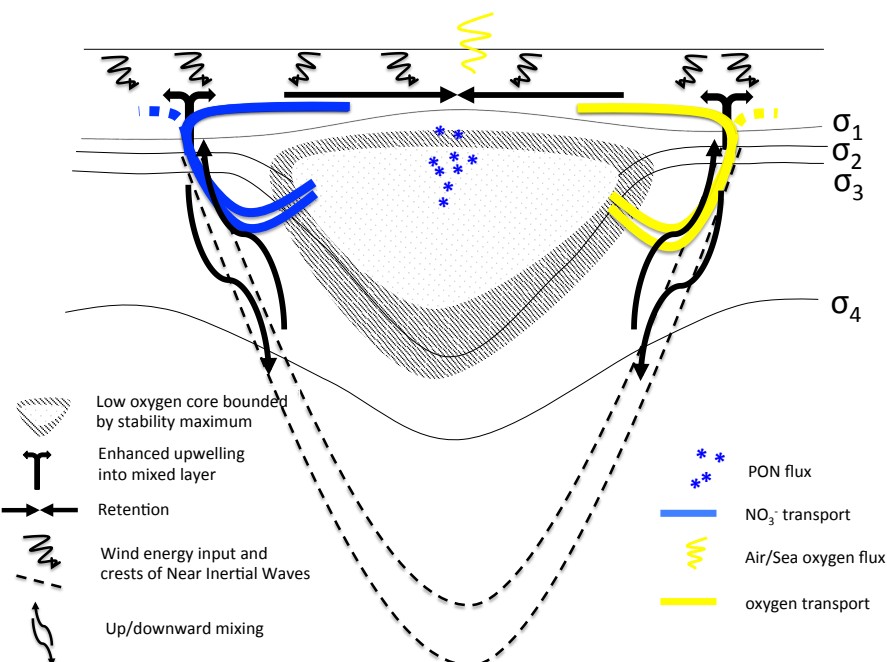

**Figure 7: Conceptual view of the physical and biogeochemical processes responsible for creating a low oxygen ACME. The recycling of Nitrate is decoupled from the oxygen cycling through the particulate matter phase (PON). The transport at the flanks and isolation of the eddy core are linked to the vertical stability ($N^2$) and the downward energy propagation of near inertial internal waves, which in turn drive enhanced mixing and thus vertical flux (upwelling and downwelling) of nutrient rich/oxygen low waters at boundaries of the eddy. The retention process ensures that part of the upwelled waters are trapped in the euphotic zone of the eddy and used for productivity. The separation (left/right) of oxygen (yellow) and $NO_3^-$ (blue) fluxes is for clarity reasons.**