# Peer review of "Upwelling and isolation in oxygen-depleted anticyclonic modewater eddies and implications for nitrate cycling"

_Biogeosciences, 2016_

## Referee Comment (RC1) · Anonymous Referee #1 · 13 Apr 2016

**1   General Comments**

The present article presents a series of observational surveys relating the existence of an oxygen-deprived mesoscale eddy core in the North Atlantic to near-inertial wave dynamics and (maybe) large-scale Ekman transport. A sequence of observations and hypotheses are suggested to account for the fact that the eddy is mostly isolated from the outside waters, but not quite. I'm actually still confused about what stays in the eddy and what gets in and out, but amendments to the articles should remedy it. At least that's my take on it is but, but I am just a physical oceanographer and I don't spend much of my time thinking about biogeochemistry.

[Figure]

In general, the processing is well done, and the graphic depictions and the accompanying text show convincing signals, raising interesting scientific questions. I would be very happy if the authors left it at that, and maybe tried their hand at process guessing in a discussion section, with larger error bars around their allegations. But in my opinion, they stretch the interpretation of their data way too far about how things are fluxed in and out of the eddy (or not), and how it explains the property structure inside of it. As far as I understand the article, they just see very interesting patterns, but are not able to prove many pieces of their model anyway. Either they are wrong, in which case this piece of text will fall into oblivion (although fig. 7 might unintentionally enjoy some form of posterity), or they are right, and the credit will go to whoever is able to prove this mechanism. Either way, I don't think they'll get citations for that part of the text. And I don't think that the article needs that to be publishable. Unless this model heavily relies on data published in other articles of their series, in which case they should consider publishing a separate article, because no-one has the time to read a whole series.

Considering that 12 co-authors could have proof-read it, the number of typos and English mistakes is rather large, even for non-native speakers. Not being a native English speaker myself, I have to let the editorial staff to correct these mistakes, but I have a list of my own if needed. Quite often, the authors prefer to use common words rather than field-specific terms ('normal eddy', 'erosion'), which would be fine if it didn't lead to ambiguities.

**2   Specific Comments:**

I will now switch to 'you' when referring to the authors.

1. P01L32: you and I seem to disagree on the specific definition of the submesoscale range. Some authors have it ranging from 1-10 km

(10.1029/177GM04), some others have it ranging from 1-50 km or even 1-100 km (10.1038/ncomms7862), but everyone seems to agree on a key value of 10 km at mid-latitudes, and Ro, Ri = O(1) in general (which is perhaps the universally accepted definition). I'm fairly confident when I say that 1 km as an upper bound is too low, and 10 meters is too small, by a long shot. There has to be some influence of the Coriolis force, that I'm certain of.

2. 1st paragraph of the intro: I'm not sure how useful this paragraph is.

3. P09, last paragraph (continued P10): I don't understand this. Why would the accumulation of NIW energy in high-N environments around an eddy shield it from mixing? If you accumulate NIWs anywhere, they tend to break, and bring mixing right at the door of the core. It sounds like planting wasp nests around one's house to prevent a wasp invasion. The whole article is confusing actually. I didn't understand it until way after, when you showed fig. 7.

4. P10L4-12: I am not sure what this paragraph is about. My take on it: does mixing work differently for nutrients than it does for other quantities? But I'm still unsure of the answer.

5. P10L22-29: A bit of ray tracing would not add much work, and could greatly improve the credibility of your hypothesis.

6. P11L14-26: my take from this paragraph: there is now an exchange pathway between the mixed layer and the core. Then what about everything you said in the preceding paragraphs? Is there a contradiction or is this a different issue?

7. P12L29-P13L15: Same problem as above. I don't find this paragraph very convincing. It is an interesting scenario, but fig. 7 is not substantiated by diffusive fluxes measurement/estimates. If Beal 2007 actually has something to say about it, you might want to use her article more, not cite her in passing. My suggestion

is that this part be moved to the discussion section, with a much more honest depiction of how little you know about why some properties are exchanged, and why some others aren't, and with a much more measured use of process-based interpretations (at least for the physical processes; I can't judge the chemistry part).

8. P11L11-15: I thought I knew what flux was until I read these sentences. What do you mean by flux? Advective flux, diffusive flux? What do you mean by erosion? What does the phrase 'NO3-/oxygen from the eddy core is primarily outward' mean? Why would a flux necessarily transfer stuff from the outside? Are you talking about a mass flux, which in all rigour should be advective? Or a diffusive flux, in which case you may or may not be right depending on the concentration distribution? And what non-dimensional number quantifies the statement 'erosion rather than flux'?

9. P11L27-32 and figure 5b: are this paragraph and figure the only ones that actually lay out your case for an influence of Ekman transport on the ACME? If so, it is a very weak case, not enough to make it to the body of the article in my opinion, and certainly not enough to make it to the abstract, Once again, it could make it to the discussion section, in passing. Thomas 2005 considers a wildly different parameter regime by the way, I don't see how it can help you support your case without more calculations.

10. P09L13: Could the low oxygen property have originated from the coast and simply have been transported all the way to here? I know that you report a decrease from 8 to 3 micromoles/grams over the course of the experiment, but I don't know the error bars on these measurements. And as far as I can tell, you simply say at some point in the text that the signal looks real or something, but that's not quite the quantified statement, especially since so much hinges on it.

11. P14L23-25: 'The NIW concept (. . .) numerical models': it depends on which models you're talking about. Numerical process studies could resolve these sorts of scales (for a low-res version of what is achievable, see 10.1175/JPO-D-14-0097.1; I am not an author, and I am not suggesting that you cite it), and could be the most obvious types of studies that could substantiate the viability of your hypotheses. So, I'd like this sentence to be rephrased in order to sound less like 'mission accomplished' and more like 'idealised process studies are needed'.

**3  Technical Comments:**

1. P01L14: extending from about 60 to 200 m depth and. . .?

2. P01L21: possibly

3. P02L03: 'has been conducted' => 'were conducted by Chaigneau. . .'

4. P02L10-13: you are describing a vertical stacking, or a baroclinic structure. Took me a while to figure out that it wasn't a radial shielding structure. And what do you mean by 'normal'? Surface-intensified or barotropic? I don't see why one is more normal than the other anyway. I would also talk about ACEs rather than AEs, to be in line with ACMEs. And can't there be CMEs?

5. P02L26-29: something odd in that sentence. Perhaps the wrong verb ('explains') is used, or a comma is missing between 'ACME' and 'with', but something is odd.

6. P03L04: 'Mesoscale eddies often have $Ro$ close to 1' => 'Although usually characterised by Ro « 1, mesoscale eddies often feature local values of $Ro$ closer to one'. See my Special Comment 1 though: you might disagree with me.

7. P03L25-26: 'the modelled . . . eddy core.' If that's the message of the paragraph, it should be placed at the beginning.

8. P03L29: by rim, do you mean top/bottom or lateral rim or both? I would say edge or boundary actually. Rim sounds like lateral boundaries, which is what you might be referring to.

9. P04L15: 'and that' => 'which'

10. P04L26: 'but purely opportunistic': huh? I think you can delete anyway, no one is judging.

11. P06L10: SA => $S_A$

12. P07L16-17: 'During the last survey... 120 m': I actually see two minima, both at 120 m. Do you mean in the vertical again?

13. P08L06: I don't see how the spiciness section shows the contrasting impact of Theta and $S_A$ on isopycnals. I don't see $\Theta$ at all actually, and I don't remember the definition of spice.

14. P08L13: 'but separating the eddy surrounding water from...' => 'but well separates the eddy core from the surrounding waters'.

15. P08L16: in the stability ratio, what is the $z$ index supposed to mean? Besides, you mix up $\theta$ and $\Theta$ here and in subsequent lines.

16. P09L09: 'but for the deeper levels more' => 'but more for the deeper levels'?

17. P09L30: 'downward also': missing word in-between?

18. P09L31: a word on what a typical AE stratification is?

19. P09L34: 'and that also' => 'which also'

20. P10L4-5: 'Having explained the isolation as..., it is tempting to...'

21. P10L06: what do you mean, 'concept'? conceptual model?

22. P11L14: 'Only vertical propagation of internal waves does not generate mixing, but (. . .)' => 'Vertical propagation of internal waves by itself does not generate mixing. In order to do so, . . .'

23. P11L15: I find it hard to conceive critical layer absorption not followed by KH.

24. P11L19-20: 'Here the mean . . . vertical mixing': I don't understand this sentence.

25. P12L25: Is the double minus in NO3- intentional?

26. P13L04: what's PON?

27. Fig. 7: A few of my colleagues (not in this field) and I unanimously agree: this figure looks too much like a particular piece of anatomy. We all suggest that you change the aspect ratio, make it less symmetric, and/or replace the blue and yellow lines by different lines. Once seen, it can't be unseen.

    Besides that, I thought oxygen was not transported in an out of the eddy (P14L15), so what's up with the yellow lines? I'd also like to see arrowheads on the blue and yellow lines, even if bi-directional (I don't think they would be). Finally, I'd like to see the huge converging arrows towards the centre of the eddy removed. I get it that some stuff is retained inside the eddy, but let's not forget that in a vortex, geostrophic or not, velocities are mostly azimuthal. I understand that this is meant to reinforce your point, but in the end, it is misleading. Or make them squiggly, which would evoke diffusion.

---

## Referee Comment (RC2) · Anonymous Referee #1 · 14 Apr 2016

I am having remorse about my last comment in the review. Since I wrote it, other people didn't see what I saw, and disagreed with me. Feel free to ignore.
* * *

---

## Short Comment (SC1) · 15 Apr 2016

I appreciate very much your comments - also the last one. I will think about how to modify the figure (without much hope in being able to substantially change it).

---

## Referee Comment (RC3) · Anonymous Referee #2 · 22 Jun 2016

Published: 11 March 2016

Review of Karstensen et al. : Upwelling and isolation in oxygen-depleted anticyclonic modewater eddies and implications for nitrate cycling.

Special issue: Hydrography, biogeochemistry, and biology of "dead-zone eddies" in the eastern tropical North Atlantic.

GENERAL COMMENTS

This work is a contribution to a special issue about "dead-zone eddies" in the Eastern North Atlantic (ETNA) where 6 manuscripts are currently available, 3 already reviewed and published in BG and the rest in discussion form.

To be concise I consider Karstensen et al. (BGD, 2016) needs MAJOR REVISION, the reasons are exposed below. My main concern about this work is the lack of a clear focus on the hypothesis, results and discussion, is it about chemical or physical oceanography?. Another important consideration is that I needed to read carefully four manuscripts within the special issue to deeply understand the results and the discussion, the manuscript (ms) is full of typos or miss-references to the figures. It seems that the authors did not check the ms coherence before submitting, this is a very bad point for their reputation. Considering the amount of coauthors an effort should have been done to ease the reading of the ms and make it a stand- alone work.

Despite this I think the ms merits to be published after some improvements both in content and layout. I understand that it is somehow difficult to organize the wealth amount of data recorded by the different surveys and observing platforms deployed to characterize this intriguing new dead zones in the ETNA.

In addition this paper is mostly about physical oceanography, and I am a chemical oceanographer, maybe the ms needs a third opinion.

A fundamental issue is the prime hypothesis of this ms which is finally resolved in Fig.7, the authors propose a physical mechanism to explain the isolation of the eddy core but also another one (near inertial waves, NIW, breaking) to explain the flux of nutrients to the upper mixed layer. As the authors say in the text the evidences to support the physical mechanisms suffer from "not having concurrent hydrography and currents data and limited options for estimating balances" (P14, L3-4). On the biogeochemical side, the authors only support their " nitrogen cycling" hypothesis with

nitrate and oxygen data from the glider surveys, but other measurements are available from typical CTD casts as described in Fiedler et al. (BGD 2016).

SPECIFIC COMMENTS

1. Introduction

Although the intro is rather long, just the last three lines contain some references to the other ms related to the studied Anticyclonic Mode Water Eddies (ACME) within the same project and using the same observing platforms. I think a comprehensive summary of the different genomic, biological and biogeochemical aspects of the ACMEs should be given, also highlighting the contribution of the current ms.

2. Data and methods

2.1. Glider survey

Maybe a word or reference about the interpolation method for the glider data would be interesting.

2.2 Glider sensor calibration

Page5, line 16. I would like to see some number about oxygen precision and accuracy, as done for nitrate (P6, L7-8). Although more details about this are surely given in Hahn et al 2014, please consider my demand.

2.3. Ship survey

I do not understand why not using the biogeochemical data gathered during M105, at least NO3, PO4, O2, particulate and dissolved organic matter, to sustain your biogeochemical interpretation of the results. More comments about this issue will be given in the corresponding section of the ms.

3. Results and Discussion

3.1 Vertical Eddy Structure

Biogeosciences is not "Journal of Physical Oceanography" so my excuses for not understanding all the difficult terms in this section. As the aim of the ms is explaining the "fluxes of nitrate" into the mixed layer supporting the high primary production in the ACME, my opinion is that an effort should be done to make the ms more readable for the ocean biogeochemical community.

P9L5-9. I checked (I read) Fiedler et al 2016 and I did not find any explanation about the translational velocity of the ACME, I found this information in Karstensen et al (BG 2015).

3.2 Eddy core isolation and vertical fluxes.

Please check the figure references in this section, it is a mess!!

It was very hard to follow the result description and the final message to be conveyed.

P9-L13 no reference to limnic systems is given in Karstensen et al (2015).

P9-L19: the nonlinearity parameter is not defined or commented previously in this work but in Karstensen et al (2015). Please explain why alpha is important for the coherence of the eddy but it does not matter to explain isolation.

P10-L2-3. Weird phrase.

P11. A mess with the figure references. Please just for the biogeochemist summarize where would NIW brake and induce mixing / fluxes in the eddy structure.

P11-L8-9. "no concurrent velocity and stratification section data exists" I do not understand, you have velocity and CTD casts from the ship so at least you have 8 stations.

3.3 Nutrient budget.

This section should be entitled "nitrate budget"... but not even so... as no budget is estimated, a better title would be "nitrate cycling" .

My main concern about this section the rejection of using other biogeochemical data from the ship surveys within the ACMEs. For example why not using the M105 NO3 and AOU data in Fig 6c?, they crossed the eddy center as showed in Fig 2b.

An evidence of denitrification would be a differential NO3:PO4 ratio.

After reading several times this section, the main question is how are the nutrients inyected into the mixed layer to support primary production?. However no profile of chlorophyll is given (I found some info about this in Loscher et al. BG 2015) , I wonder if the gliders have at least a backscattering or fluorometer sensor.

The biogeochemical info in Fiedler et al BGD 2016 in the shelf, CVOO and the eddies may help to explain the high primary production (PP), if eddies are formed in the shelf, they contain nutrients that are used and converted into organic matter (particulate and dissolved ) that sinks and is remineralized in the eddy creating the O2 minimum. Is it enough the initial NO3 in the shelf to sustain PP in the eddy when it moves into the ETNA?. Does it really need an extra NO3 input?.

It is very hard to understand a decoupled O2 and NO3 cycle if denitrification is not important. Please check the NO3:PO4 ratio. An anomalous O2:NO3 ratio could be related to the stochiometry of the organic matter remineralized both particulate and dissolved, please check the available data.

4. Summary and conclusions

I suppose it would need to be rewritten depending on the results from section 3.3.

I hope to have been helpful.

---

## Author Comment (AC1) · 23 Oct 2016

**Detailed response to reviewers comments**

We thank both the reviewers for encouraging but also critical words. We have revised and restructured the text. We hope that the new version is not only better synthesizing our results but further addresses adequately the points were improvements have been suggested by the reviewers.

**Anonymous Referee #1**

**General Comments**

The present article presents a series of observational surveys relating the existence of an oxygen-deprived mesoscale eddy core in the North Atlantic to near-inertial wave dynamics and (maybe) large-scale Ekman transport. A sequence of observations and hypotheses are suggested to account for the fact that the eddy is mostly isolated from the outside waters, but not quite. I'm actually still confused about what stays in the eddy and what gets in and out, but amendments to the articles should remedy it. At least that's my take on it is but, but I am just a physical oceanographer and I don't spend much of my time thinking about biogeochemistry.

In general, the processing is well done, and the graphic depictions and the accompanying text show convincing signals, raising interesting scientific questions. I would be very happy if the authors left it at that, and maybe tried their hand at process guessing in a discussion section, with larger error bars around their allegations. But in my opinion, they stretch the interpretation of their data way too far about how things are fluxed in and out of the eddy (or not), and how it explains the property structure inside of it. As far as I understand the article, they just see very interesting patterns, but are not able to prove many pieces of their model anyway. Either they are wrong, in which case this piece of text will fall into oblivion (although fig. 7 might unintentionally enjoy some form of posterity), or they are right, and the credit will go to whoever is able to prove this mechanism. Either way, I don't think they'll get citations for that part of the text. And I don't think that the article needs that to be publishable. Unless this model heavily relies on data published in other articles of their series, in which case they should consider publishing a separate article, because no-one has the time to read a whole series.

Considering that 12 co-authors could have proof-read it, the number of typos and English mistakes is rather large, even for non-native speakers. Not being a native English speaker myself, I have to let the editorial staff to correct these mistakes, but I have a list of my own if needed. Quite often, the authors prefer to use common words rather than field-specific terms

('normal eddy', 'erosion'), which would be fine if it didn't lead to ambiguities.

We thank the reviewer for the detailed and very useful comments. Based on the reviewer comments we hope that we were able to better (as far as it is possible based on the data at hand) discuss the physical processes that are at work in the eddy.

Based on the data at hand we can describe the stratification, currents, and biogeochemical characteristic of the eddy, and also some temporal evolution. We refer to results published elsewhere in order to interpret what we observe. The paper by Sheen et al. (2015) that describes the Near Inertial Wave (NIW) propagation in and around a Modewater eddy (deep Southern Ocean eddy) based on observational (microstructure) data. Sheen et al. found by applying a ray trace model to the observed N2 profiles, that the core of the eddy does allow only a selected range of incidence NIW to enter. All other NIW were reflected at the periphery of the core at the N2 maximum and wave/wave interaction was suggested to generate observed enhanced mixing. One part which was misinterpreted in the last version was related to the NIW propagation in regions where $f_{eff}<f$. Indeed NIW can propagate in region with $f_{eff}<f$ such as the core the anticyclone. However, more relevant is the region just outside the eddy and were the horizontal velocity shear generates $f_{eff}>f$ (e.g. Halle and Pinkel 2003; Fig. 16). Here, NIW generated in an f-region are forced to propagate downward to depth. Enhanced mixing by shear instabilities from NIW currents that periodically enhance the background flow have been reported (Kawaguchi et al. 2016).

The comments about the quality of the writing are fully to the account of the lead author. In fact, the Guest Editor had kindly provided a proofread version that could have been used for initial publication – but unfortunately the file was "overlooked" by the lead author in the submission process. All comments have been considered in the revised version.

**Specific Comments:**

I will now switch to 'you' when referring to the authors.

1. P01L32: you and I seem to disagree on the specific definition of the submesoscale range. Some authors have it ranging from 1-10 km (10.1029/177GM04), some others have it ranging from 1-50 km or even 1-100 km (10.1038/ncomms7862), but everyone seems to agree on a key value of 10 km at mid-latitudes, and Ro, Ri = O(1) in general (which is perhaps the universally accepted definition). I'm fairly confident when I say that 1 km as an upper bound is too low, and 10 meters is too small, by a long shot. There has to be some influence of the Coriolis force, that I'm certain of.

A very valid comment – for the submesoscale range we followed the recent definition given by McWilliams (2016): "To be more quantitative, the approximate scale ranges for SMCs (*submesoscale currents*) are l=0.1–10km in the horizontal, h=0.01–1 km in the vertical, and hours-days in time (except for some submesoscale coherent vortices (SCVs) that can wander around in the vertical interior with lifetimes of years)."

**1st paragraph of the intro: I'm not sure how useful this paragraph is.**

This is true - we have shortened the paragraph, omitted the eddy detection sentences and restructured the paragraph.

P09, last paragraph (continued P10): I don't understand this. Why would the accumulation of NIW energy in high-N environments around an eddy shield it from mixing? If you accumulate NIWs anywhere, they tend to break, and bring mixing right at the door of the core. It sounds like planting wasp nests around one's house to prevent a wasp invasion. The whole article is confusing actually. I didn't understand it until way after, when you showed fig. 7.

We are sorry for the confusion. We take from this comment that the reviewer finally (fig. 7) understood the mechanism but not in this paragraph were it was described. As a consequence we re-wrote the paragraph (but also the introduction paragraph on lowering/increasing f around anticyclonic eddies and the impact on the propagation of NIW). It is also of important to mention that we wrongly interpreted the $f_{eff}$ pattern. It is in fact not the lowering of the planetary vorticity in the core of ACME/AC but the increase in $f_{eff}$ at the transition zone between the eddy and the surrounding waters that forces NIW to propagate downward and eventually cause mixing (see e.g. Halle and Pinkel 2003; Fig. 16). This correction also required some modification on figure 7 – which might be appreciated by this particular reviewer mentioning some concerns with the graphical realisation.

P10L4-12: I am not sure what this paragraph is about. My take on it: does mixing work differently for nutrients than it does for other quantities? But I'm still unsure of the answer.

Our intention was to discuss differences in surface signatures of nutrient upwelling (primary productivity) – is it more at the edge of an eddy or in the centre? The paragraph did not consider other quantities. However, we re-wrote the paragraph.

P10L22-29: A bit of ray tracing would not add much work, and could greatly improve the credibility of your hypothesis.

We do not see reasons to question the applicability of the Sheen et al. (2015) ray tracing to the ACME we observe. The ray tracing is based on an N2 profiles only and the Sheen at all and our N2 look very similar. Moreover, enhanced mixing at the N2 maximum in an ACME was also recently reported for an Arctic eddy (Kawaguchi et al. 2016) and that further support our interpretation of the data. What we actually miss in our observations is microstructure data that would help to quantify the mixing efficiency across an ACME.

P11L14-26: my take from this paragraph: there is now an exchange pathway between the

mixed layer and the core. Then what about everything you said in the preceding paragraphs? Is there a contradiction or is this a different issue?

The exchange is focussed at the rim or edge – here is were we observe the NIW to propagate downward. The NIW also propagate "outward" from the N2 maximum (see Sheen et al. 2015, Kawaguchi et al. 2016). There is no evidence from our data that support an exchange of the core with the surroundings. The term "erosion" should emphasize that the mixing is just on one side of the eddy - "outward" from the N2 maximum" and the core properties are largely unaffected. The term erosion has been used in the past in describing process that operate at the edge of warm core eddies (citation: "note that lateral intrusion and mixing on the sides of the eddy are contributing most to its erosion" Kroll, 1993).

P12L29-P13L15: Same problem as above. I don't find this paragraph very convincing. It is an interesting scenario, but fig. 7 is not substantiated by diffusive fluxes measurement/estimates. If Beal 2007 actually has something to say about it, you might want to use her article more, not cite her in passing. My suggestion is that this part be moved to the discussion section, with a much more honest depiction of how little you know about why some properties are exchanged, and why some others aren't, and with a much more measured use of process-based interpretations (at least for the physical processes; I can't judge the chemistry part).

Indeed we can argue only based in what has been reported by others on mixing in ACMEs (shallow ACME: Kawaguchi et al. 2016); deep ACME (Sheen et al. 2015). We followed your advice and move this part to the end of the paper.

P11L11-15: I thought I knew what flux was until I read these sentences. What do you mean by flux? Advective flux, diffusive flux? What do you mean by erosion? What does the phrase 'NO3-/oxygen from the eddy core is primarily outward' mean? Why would a flux necessarily transfer stuff from the outside? Are you talking about a mass flux, which in all rigour should be advective? Or a diffusive flux, in which case you may or may not be right depending on the concentration distribution? And what non-dimensional number quantifies the statement 'erosion rather than flux'?

We suspect you mean P13L11—15? The problem with a gradient flux considering an advective/ diffusive balance is that it would EXCHANGE properties – hence the core would be altered in its properties (e.g. Sa / Theta). What we actually observe is a remarkable constant T/S (and a decrease in oxygen over time – see figure below). The observations of a maximum in mixing efficiency at the N2 maximum by Sheen et al. (2015) and Kawaguchi et al. (2016), combined with the minimum in mixing efficiency in the core of the ACME (that is in-line with the NIW propagation pathways as simulated by Sheen et al. 2015) support an "erosion" scenario. With

"erosion" we mean a "shrinking" of the ACME core. We modified the text in order to make this point more clear. The TS diagram may further help – it shows the eddy core profiles from the two glider surveys. What can be seen is that the TS in the core is very stable but that mixing at the edges has "eroded" or shrinken the core.

[Figure]

P11L27-32 and figure 5b: are this paragraph and figure the only ones that actually lay out your case for an influence of Ekman transport on the ACME? If so, it is a very weak case, not enough to make it to the body of the article in my opinion, and certainly not enough to make it to the abstract, Once again, it could make it to the discussion section, in passing. Thomas 2005 considers a wildly different parameter regime by the way, I don't see how it can help you support your case without more calculations.

We agree and removed the paragraph.

P09L13: Could the low oxygen property have originated from the coast and simply have been transported all the way to here? I know that you report a decrease from 8 to 3 micromoles/grams over the course of the experiment, but I don't know the error bars on these measurements. And as far as I can tell, you simply say at some point in the text that the signal looks real or something, but that's not quite the quantified statement, especially since so much hinges on it.

A very valid comment! We could show in the past (Karstensen et al. 2015, Fiedler et al. 2016, Schütte et al. 2016) that the low oxygen core did not originate from the coast. For example, direct observation of an Argo float with oxygen sensor that was trapped in a CE over a period of more than 7 month (Karstensen et al. 2015) from the upwelling region into the open North Atlantic showed a constant decrease in oxygen in the eddy core. Also from a number of direct observations of eddies that were surveyed shortly after they detached from the coast and many month later again (Karstensen et al. 2015, Fiedler et al. 2016, Schütte et al. 2016).

P14L23-25: 'The NIW concept (. . .) numerical models': it depends on which models you're talking about. Numerical process studies could resolve these sorts of scales (for a low-res version of what is achievable, see 10.1175/JPO-D-14- 0097.1; I am not an author, and I am not suggesting that you cite it), and could be the most obvious types of studies that could substantiate the viability of your hypotheses. So, I'd like this sentence to be rephrased in order to sound less like 'mission accomplished' and more like 'idealised process studies are needed'

Thank you for the comment. Of course there are models that do resolve the scales and hopefully the processes. We rephrased the sentence accordingly.

**Technical Comments:**

P01L14: extending from about 60 to 200 m depth and. . .?  - done

P01L21: possibly  -done

P02L03: 'has been conducted' => 'were conducted by Chaigneau. . .'  - sentence removed

P02L10-13: you are describing a vertical stacking, or a baroclinic structure. Took me a while to figure out that it wasn't a radial shielding structure. And what do you mean by 'normal'? Surface-intensified or barotropic? I don't see why one is more normal than the other anyway. I would also talk about ACEs rather than AEs, to be in line with ACMEs. And can't there be CMEs? – We rephrased the sentence and hope it is now clear that describe the stratification and the Mode. In the context of water masses the word "Mode" is often used for nearly homogenous properties such as for subtropical, or subpolar Mode Waters. We are not aware of a publication on "Cyclonic Modewater eddies" but would be happy to add a reference if the reviewer could provide one.

P02L26-29: something odd in that sentence. Perhaps the wrong verb ('explains') is used, or a comma is missing between 'ACME' and 'with', but something is odd.  - we rephrased the

sentence.

P03L04: 'Mesoscale eddies often have Ro close to 1' => 'Although usually characterised by Ro «
1, mesoscale eddies often feature local values of Ro closer to one'. See my Special Comment 1
though: you might disagree with me. – We rephrased the sentence accordingly.

P03L25-26: 'the modelled . . . eddy core.' If that's the message of the paragraph, it should be
placed at the beginning.  – We rephrased the whole paragraph.

P03L29: by rim, do you mean top/bottom or lateral rim or both? I would say edge or boundary
actually. Rim sounds like lateral boundaries, which is what you might be referring to.  – We
rephrased the sentence

P04L15: 'and that' => 'which'   – We rephrased the whole paragraph.

P04L26: 'but purely opportunistic': huh? I think you can delete anyway, no one is judging.  - was
deleted

P06L10: SA => $S_A$  -changed

P07L16-17: 'During the last survey. . . 120 m': I actually see two minima, both at  120 m. Do you
mean in the vertical again?   - The sentence referred to the vertical and we modified the
sentence.

P08L06: I don't see how the spiciness section shows the contrasting impact of Theta and $S_A$ on
isopycnals. I don't see Θ at all actually,
and I don't remember the definition of
spice.  -Spiciness is constructed to be a
variable that is most sensitive to isopycnal
thermohaline variations, and least
correlated with the density field (Flambert
2002). Because both, Theta and SA,
contribute to density it is redundant to
analyse them jointly along isopycnals. We
now added the definition for spiciness to
the text (not the equation). The reason
why Theta is not shown is because not
much can be learned from the section –
but if the reviewer insists we could add it.

[Figure]

P08L13: 'but separating the eddy surrounding water from. . .' => 'but well separates the eddy core from the surrounding waters'. –we rephrased  the sentence

P08L16: in the stability ratio, what is the z index supposed to mean? Besides, you mix up θ and Θ here and in subsequent lines.  –It is the vertical gradient/contribution. An explantion for z was now added to the text. Thank you for mentioning the problem with θ and Θ which is related to using the word Equation editor or the symbol set.

P09L09: 'but for the deeper levels more' => 'but more for the deeper levels'?  - Changed accordingly

P09L30: 'downward also': missing word in-between?  - Changed sentence

P09L31: a word on what a typical AE stratification is?   - Sentence was removed (whole section rephrased)

P09L34: 'and that also' => 'which also'   - Changed sentence

P10L4-5: 'Having explained the isolation as..., it is tempting to...'   - Sentence was removed (whole section rephrased)

P10L06: what do you mean, 'concept'? conceptual model?   - Sentence was removed (whole section rephrased)

P11L14: 'Only vertical propagation of internal waves does not generate mixing, but (...)' => 'Vertical propagation of internal waves by itself does not generate mixing. In order to do so, . . .'
 - Changed sentence

P11L15: I find it hard to conceive critical layer absorption not followed by KH.    - That is true and we changed the sentence

P11L19-20: 'Here the mean . . . vertical mixing': I don't understand this sentence.  -Changed sentence: "Here the mean flow could gain energy from the NIW current that in turn could lead to energy dissipation because of the shear-instability (Kawaguchi et al. 2016)"

P12L25: Is the double minus in NO3- intentional?  -Typo, Changed

P13L04: what's PON? – It stands for "Particulate Organic Nitrogen"  (which is nitrogen that is part of particles made out of organic substances)

Fig. 7: A few of my colleagues (not in this field) and I unanimously agree: this figure looks too

much like a particular piece of anatomy. We all suggest that you change the aspect ratio, make it less symmetric, and/or replace the blue and yellow lines by different lines. Once seen, it can't be unseen. Besides that, I thought oxygen was not transported in an out of the eddy (P14L15), so what's up with the yellow lines? I'd also like to see arrowheads on the blue and yellow lines, even if bi-directional (I don't think they would be). Finally, I'd like to see the huge converging arrows towards the centre of the eddy removed. I get it that some stuff is retained inside the eddy, but let's not forget that in a vortex, geostrophic or not, velocities are mostly azimuthal. I understand that this is meant to reinforce your point, but in the end, it is misleading. Or make them squiggly, which would evoke diffusion. - We discussed that online and some modification to the figure was needed.

**Reference:**

McWilliams JC. 2016 Submesoscale currents in the ocean. Proc. R. Soc. A 472: 20160117. http://dx.doi.org/10.1098/rspa.2016.0117

Halle, C., and R. Pinkel: Internal wave variability in the Beaufort Sea during the winter of 1993/1994, J. Geophys. Res., 108(C7), 3210, doi:10.1029/2000JC000703, 2003.

Kawaguchi, Y., S. Nishino, J. Inoue, K. Maeno, H. Takeda, and K. Oshima: Enhanced diapycnal mixing due to near-inertial internal waves propagating through an anticyclonic eddy in the ice-free Chukchi Plateau. J. Phys. Oceanogr., 46, 2457-2481, doi:10.1175/JPO-D-15-0150.1, 2016.

Flament, P: A state variable for characterizing water masses and their diffusive stability: spiciness, Progress in Oceanography 54, 493–501, 2002

**Anonymous Referee #2**

**GENERAL COMMENTS**

This work is a contribution to a special issue about "dead-zone eddies" in the Eastern North Atlantic (ETNA) where 6 manuscripts are currently available, 3 already reviewed and published in BG and the rest in discussion form.

To be concise I consider Karstensen et al. (BGD, 2016) needs MAJOR REVISION, the reasons are exposed below. My main concern about this work is the lack of a clear focus on the hypothesis, results and discussion, is it about chemical or physical oceanography?

Another important consideration is that I needed to read carefully four manuscripts within the special issue to deeply understand the results and the discussion, the manuscript (ms) is full of typos or miss- references to the figures. It seems that the authors did not check the ms coherence before submitting, this is a very bad point for their reputation. Considering the amount of coauthors an effort should have been done to ease the reading of the ms and make it a stand- alone work.

Despite this I think the ms merits to be published after some improvements both in content and layout. I understand that it is somehow difficult to organize the wealth amount of data recorded by the different surveys and observing platforms deployed to characterize this intriguing new dead zones in the ETNA. In addition this paper is mostly about physical oceanography, and I am a chemical oceanographer, maybe the ms needs a third opinion.

We thank the reviewer for encouraging but also critical words. Indeed it is a problem to publish papers that address the interaction of physical and biogeochemical processes. To our opinion the Guest editor did a very good job in selecting reviewers that, although addressing primarily points related to their own discipline (Reviewer 1 being a physical oceanographer, Reviewer 2 being a chemical Oceanographer), asked apparently "simple questions" on others discipline which are often the trickiest to answer. As Reviewer 2 will see, we have revised the text and reordered the content over many parts. We hope that the new version is now focussing to the point and thus is easier to read. We tried to simplify the physical and biogeochemical parts. However, certain parts might be difficult to fully understand by one or the other disciplinary reader.

Indeed the many typos in the submitted version go fully to the account of the lead author. In fact, the Guest editor had kindly provided a proofread version that could have been used for publication – but unfortunately the file was "overlooked" by the lead author in the submission process.

A fundamental issue is the prime hypothesis of this ms which is finally resolved in Fig.7, the authors propose a physical mechanism to explain the isolation of the eddy core but also another one (near inertial waves, NIW, breaking) to explain the flux of nutrients to the upper mixed layer. As the authors say in the text the evidences to support the physical mechanisms suffer from "not having concurrent hydrography and currents data and limited options for

estimating balances" (P14, L3-4). On the biogeochemical side, the authors only support their "nitrogen cycling" hypothesis with nitrate and oxygen data from the glider surveys, but other measurements are available from typical CTD casts as described in Fiedler et al. (BGD 2016).

[Figure]

A new publication in the physical mechanisms of mixing related to the ACME has been published in the meantime (Kawaguchi et al. 2016) that further supports our mixing/erosion of the eddy core. Of course we do not have other data at hand. However, we added one figure that nicely shows the high particle load that "rains" into the eddy core and which further supports the idea that particle sinking and remineralization is one key process in creating the low oxygen core. Fiedler et al. (2016) do not present AOU/NO3 ratios but, the data

**SPECIFIC COMMENTS**

Introduction
Although the intro is rather long, just the last three lines contain some references to the other ms related to the studied Anticyclonic Mode Water Eddies (ACME) within the same project and using the same observing platforms. I think a comprehensive summary of the different genomic, biological and biogeochemical aspects of the ACMEs should be given, also highlighting the contribution of the current ms.

We re-wrote and restructured the introduction. In particular the eddy detection part was removed (irrelevant for the present study). Details about the different genomic, biological and biogeochemical aspects of the particular ACME that we investigate here will be given in an overview article for the special issue (In preparation).

2. Data and methods
2.1. Glider survey
Maybe a word or reference about the interpolation method for the glider data would be interesting.
A linear interpolation was applied (now added to the text)

2.2 Glider sensor calibration
Page5, line 16. I would like to see some number about oxygen precision and accuracy, as done for nitrate (P6, L7-8). Although more details about this are surely given in Hahn et al 2014, please consider my demand.

Sorry for not providing the errors estimates. The comparison between calibrated (by titration of oxygen samples) Clarke sensor on the CTD and the calibrated optode data suggests an overall (full oxygen range) RMS error of 3 µmol kg−1. However, for the chemically forced zero oxygen an RMS error of 1 µmol kg−1 is expected. We added the information to the text.

2.3. Ship survey
I do not understand why not using the biogeochemical data gathered during M105, at least NO3, PO4, O2, particulate and dissolved organic matter, to sustain your biogeochemical interpretation of the results. More comments about this issue will be given in the corresponding section of the ms.
-As outlined under the specific points below, we primarily reference to the published figures in the accompanying articles.

3. Results and Discussion
3.1 Vertical Eddy Structure Biogeosciences is not "Journal of Physical Oceanography" so my excuses for not understanding all the difficult terms in this section. As the aim of the ms is explaining the "fluxes of nitrate" into the mixed layer supporting the high primary production in the ACME, my opinion is that an effort should be done to make the ms more readable for the ocean biogeochemical community.
-We hope that that by rewriting the manuscript over large parts has fixed this issue.

P9L5-9. I checked (I read) Fiedler et al 2016 and I did not find any explanation about the translational velocity of the ACME, I found this information in Karstensen et al (BG 2015).
In section "3.1. Eddy Characteristics" Fielder et al. discuss the translation velocity. However, numbers are given in Karstensen et al. (2015) and Schütte et al. (2016). We changed accordingly.

3.2 Eddy core isolation and vertical fluxes. Please check the figure references in this section, it is a mess!! It was very hard to follow the result description and the final message to be conveyed.

P9-L13 no reference to limnic systems is given in Karstensen et al (2015). -In the abstract of Karstensen et al. (2015) it says: "…create the "dead zone" inside the eddies, so far only reported for coastal areas or lakes." – with "lakes" we anticipated limnic systems.

P9-L19: the nonlinearity parameter is not defined or commented previously in this work but in Karstensen et al (2015). Please explain why alpha is important for the coherence of the eddy but it does not matter to explain isolation. -Eddies might be separated into linear waves (also called "Rossby Waves") or in isolated, coherent structures. The non-linearity parameter is a measure to judge if the feature is a linear wave (translation speed and rotation speed are similar) or a coherent eddy (rotation speed much higher than translation) with a different dynamical regime. We refer in the text to the isolation of the core against lateral or vertical mixing – maybe shielding for mixing is a better formulation? The paragraph has been rewritten.

P10-L2-3. Weird phrase. – Indeed, but the paragraph has been completely rewritten and the sentence is removed.

P11. A mess with the figure references. Please just for the biogeochemist summarize where would NIW brake and induce mixing / fluxes in the eddy structure.
- We rephrased the sentences and hope that we provide with Figure 7 a good overview about were exactly mixing occurs.

P11-L8-9. "no concurrent velocity and stratification section data exists" I do not understand, you have velocity and CTD casts from the ship so at least you have 8 stations. –We did that when estimating the gradient Richardson number (P11L8) but the statistical significance is very low.

3.3 Nutrient budget.
This section should be entitled "nitrate budget"... but not even so... as no budget is estimated, a better title would be "nitrate cycling" .
- This is true and we changed that accordingly

My main concern about this section the rejection of using other biogeochemical data from the ship surveys within the ACMEs. For example why not using the M105 NO3 and AOU data in Fig 6c?, they crossed the eddy center as showed in Fig 2b.
– It is less a "rejection" but the wish to avoid showing plots that have been presented by other authors already. On the left a summary of NO3:AOU from cruise data at different locations.

[Figure]

An evidence of denitrification would be a differential NO3:PO4 ratio.
- The NO3:PO4 ratio has been presented in Löscher et al. (2015) in Figure 3c. The glider survey does not provide PO4 data – as such it is unclear what the bottle NO3/PO4 figure would add to our discussion? Probably the Löscher et al. (2015) was not sufficiently cited?

After reading several times this section, the main question is how are the nutrients injected into the mixed layer to support primary production?. However no profile of chlorophyll is given (I found some info about this in Loscher et al. BG 2015) , I wonder if the gliders have at least a backscattering or fluorometer sensor.
- We added new figures on turbidity and fluorescence that hopefully now show better the particle sinking (turbidity) and the fluorescence peak in the mixed layer/at the mixed layer base.

The biogeochemical info in Fiedler et al BGD 2016 in the shelf, CVOO and the eddies may help to explain the high primary production (PP), if eddies are formed in the shelf, they contain nutrients that are used and converted into organic matter (particulate and dissolved) that sinks

and is remineralized in the eddy creating the O2 minimum. Is it enough the initial NO3 in the shelf to sustain PP in the eddy when it moves into the ETNA?. Does it really need an extra NO3 input?

- The problem is the process that refuels the euphotic zone of the eddy with nutrients. It is not a problem of the initial nutrient content in the eddy core. The process we propose (and that alters fundamentally the NO3:AOU ratio) is a recycling of nutrients, which in turn is the results of the specifics of upwelling in the eddy. We would not call that an "extra NO3" but a recycling that alters the AOU/NO3 ratio. Figure 4 in Fiedler et al. (2016) shows the profiles from the cruises as well as from the shelf. It can be seen that the shelf water are lower in NO3 (and higher in O2)

It is very hard to understand a decoupled O2 and NO3 cycle if denitrification is not important. Please check the NO3:PO4 ratio. An anomalous O2:NO3 ratio could be related to the stochiometry of the organic matter remineralized both particulate and dissolved, please check the available data.

-In Löscher et al. (2015) (and in Grundle et al. submitted) the nitrogen loss by denitrification was in the nanomolar range but the nitrate deficit is in the micromolar range. In fact the conceptual model we provide here should explain exactly this disequilibrium without the need of denitrification (conceptualized through Figure 7).

For our concept the key is that the specific mixing (created by the submesoscale dynamics in and around the eddy) is taking away ("erosion") part of the eddy core. This part has high NO3/low O2 water enters by mixing induced pathways the mixed layer/euphotic zone. However, once in the mixed layer the pathways of the high NO3 and the low O2 water are different. The low O2 water will lower the oxygen content in the mixed layer (being now undersaturated in oxygen) but which is refuelled by air/sea gas exchange. In contrast, the high NO3 water is used for PP in the mixed layer and, through PP is incorporated into particles (as PON). The particles (with the PON) sink out of the mixed layer back into the core. This some of the NO3 is re-entering the mixed layer. Essentially this is a gravitational process and O2 does not participate in it.

4. Summary and conclusions

I suppose it would need to be rewritten depending on the results from section 3.3.

-We have rewritten the section 4 and hope to made the points now more clear.

I hope to have been helpful.

-Definitely – Thank you!

---

## Author Response (AR1)

**Detailed response to guest editor and reviewers comments**

We thank the guest editor Denis Gilbert and the two anonymous reviewers for evaluating the manuscript and providing constructive comments. We have revised and restructured the text over large parts, reworked the figures, and included (as requested by reviewer #2) one new figure. We redesigned Figure 8 (former 7) substantially. We hope that the new version of the manuscript is not only better synthesizing our results but further addresses adequately the points were improvements have been suggested. In the revision of the manuscript a more detailed discussion on potential mixing regions has been added by making use of the gradient Richardson number.

We now considered the erosion signal of the core that is clearly visible (see also a redesigned Fig. 4) but ignored in the previous submission. We did not include a ray trace model using our N2 and u,v shear (the application was suggested by the Reviewer#1 and taken up by the guest editor). The ray trace model (as in Sheen et al. 2015) will not provide any detail about potential mixing. In Sheen et al. (2015) mixing observations from Microstructure probes were available and the ray tracing helped to interpret the results. However, the mechanisms for mixing in Sheen et al. (2015) are speculative only. We do not even have direct mixing observations and as such not even a qualitative comparison beyond what is already presented in Sheen et al. can be expected. Sheen et al. (2015) note about the limitation of their model: *"We note that these simple ray tracing models are presented as a heuristic tool and are not intended to capture the full range of wave-mean flow interactions at play in such a complex system. For example, the models fail to account for the breakdown of the WKB analysis at critical layers [Booker and Bretherton, 1967; Jones, 1967], the consequences of using the WKB approximation in a background flow with relatively small horizontal length scales (O(10 km)) [Olbers, 1981; Whitt and Thomas, 2012], the assumption that the mean flow is 1-D and rectilinear [Bühler and McIntyre, 2005; Polzin, 2008], the potential for loss/gain of wave energy to the mean flow and/or a wave-induced effective viscosity [Booker and Bretherton, 1967; Muller, 1976; Polzin, 2010], the effect of f /N approaching 0.1 [e.g., Gerkema and Exarchou, 2008], the influence of the eddy rotation and the inclusion of the vorticity term in the ray tracing formulation [Kunze et al., 1995], and double diffusive processes."*

We think the Ri calculations are the maximum we could do to narrow down the potential mixing regions. Further studies with dedicated mixing experiments (e.g. MSS, dye experiments) in low oxygen eddies are required to provide observational ground truth which in turn can then be validated against e.g. ray tracing models.

**Guest editors comments**

Thank you for providing your constructive comments. Please find our response to your specific comments below:

Given that the eddy core described by Sheen et al. (2015) is located at a much greater depth (about 2000 m) than the 125 m deep core of the anticyclonic mode-water eddy (ACME) of your study, I believe that ray tracing specifically done for your study's ACME would help better determine the critical layer depth(s). See Referee # 1, Specific comment # 5;

We decided to not apply a ray trace model to our data for the reasons outlined in the summary section above – we hope that you agree to our conclusion.

Is vertical mixing really taking place mostly on the lateral edges of the eddy, as shown on your Fig. 7, rather than near the top of the eddy? Figure 4c of Sheen et al. (2015) shows enhanced mixing at the top of the eddy, at about the same depth (1500 m in their case) as where their critical layer scenario is located (their Fig. 5c).

Indeed, we revised that part of the manuscript substantially. By comparing the profiles and T/S diagrams from different life-stages of the eddy the impact of the "mixing", better "erosion", is clearly seen in the profiles and has not be adequately considered in the first submission. We redesigned Figure 4 that helps to discuss the temporal evolution in greater detail.

**Anonymous Referee #1**

**General Comments**

The present article presents a series of observational surveys relating the existence of an oxygen-deprived mesoscale eddy core in the North Atlantic to near-inertial wave dynamics and (maybe) large-scale Ekman transport. A sequence of observations and hypotheses are suggested to account for the fact that the eddy is mostly isolated from the outside waters, but not quite. I'm actually still confused about what stays in the eddy and what gets in and out, but amendments to the articles should remedy it. At least that's my take on it is but, but I am just a physical oceanographer and I don't spend much of my time thinking about biogeochemistry.

In general, the processing is well done, and the graphic depictions and the accompanying text show convincing signals, raising interesting scientific questions. I would be very happy if the authors left it at that, and maybe tried their hand at process guessing in a discussion section, with larger error bars around their allegations. But in my opinion, they stretch the

interpretation of their data way too far about how things are fluxed in and out of the eddy (or not), and how it explains the property structure inside of it. As far as I understand the article, they just see very interesting patterns, but are not able to prove many pieces of their model anyway. Either they are wrong, in which case this piece of text will fall into oblivion (although fig. 7 might unintentionally enjoy some form of posterity), or they are right, and the credit will go to whoever is able to prove this mechanism. Either way, I don't think they'll get citations for that part of the text. And I don't think that the article needs that to be publishable. Unless this model heavily relies on data published in other articles of their series, in which case they should consider publishing a separate article, because no-one has the time to read a whole series.

Considering that 12 co-authors could have proof-read it, the number of typos and English mistakes is rather large, even for non-native speakers. Not being a native English speaker myself, I have to let the editorial staff to correct these mistakes, but I have a list of my own if needed. Quite often, the authors prefer to use common words rather than field-specific terms ('normal eddy', 'erosion'), which would be fine if it didn't lead to ambiguities.

We thank the reviewer for the detailed and constructive comments. Based on the reviewer comments we hope that we were able to better (as far as it is possible based on the data at hand) discuss the physical (and biogeochemical) processes that are at work in the eddy.

Based on the data at hand we can describe the stratification, currents, and biogeochemical characteristic of the eddy, and its temporal evolution. We added now a more detailed analysis of mixing based on a gradient Richardson number approach. Moreover results published elsewhere are considered to interpret what we observe. The paper by Sheen et al. (2015) describes the possible Near Inertial Wave (NIW) propagation in and around a Modewater eddy (deep Southern Ocean eddy) based on observational (microstructure) data. Halle and Pinkel (2003) concluded that NIW propagation through a low stratified core impact the NIW energy density – the NIW propagation is very much increase in the low stratified core while in parallel the energy density decreases and thus mixing is supressed.

One part which was misinterpreted in the last version of the manuscript was related to the NIW propagation in regions where $f_{eff}<f$. Indeed NIW can propagate in region with $f_{eff}<f$ (superinertial) such as for the core of an anticyclone. Here the NIW energy propagates downward. However, outside the eddy the horizontal velocity shear generates a region with $f_{eff}>f$ (e.g. Halle and Pinkel 2003; their Fig. 16). Here, NIW generated in an f-region are forced to propagate downward when entering such a region until they are eventually reflected or loose energy by dissipation (critical layer). Enhanced mixing by shear instabilities from NIW currents that periodically enhance the background flow has been reported (Kawaguchi et al. 2016).

The comments about the quality of the writing are fully to the account of the lead author. In fact, the Guest Editor had kindly provided a proofread version that could have been used for initial publication – but unfortunately the file was "overlooked" by the lead author in the

submission process.  All comments have been considered in the revised version.

**Specific Comments:**

I will now switch to 'you' when referring to the authors.

1. P01L32: you and I seem to disagree on the specific definition of the submesoscale range. Some authors have it ranging from 1-10 km (10.1029/177GM04), some others have it ranging from 1-50 km or even 1-100 km (10.1038/ncomms7862), but everyone seems to agree on a key value of 10 km at mid-latitudes, and Ro, Ri = O(1) in general (which is perhaps the universally accepted definition). I'm fairly confident when I say that 1 km as an upper bound is too low, and 10 meters is too small, by a long shot. There has to be some influence of the Coriolis force, that I'm certain of.

A very valid comment – for the submesoscale range we followed the recent definition given by McWilliams (2016): "To be more quantitative, the approximate scale ranges for SMCs (*submesoscale currents*) are l=0.1–10km in the horizontal, h=0.01–1 km in the vertical, and hours-days in time (except for some submesoscale coherent vortices (SCVs) that can wander around in the vertical interior with lifetimes of years)."

1st paragraph of the intro: I'm not sure how useful this paragraph is.

This is true - we have shortened the paragraph, omitted the eddy detection sentences and restructured the paragraph.

P09, last paragraph (continued P10): I don't understand this. Why would the accumulation of NIW energy in high-N environments around an eddy shield it from mixing? If you accumulate NIWs anywhere, they tend to break, and bring mixing right at the door of the core. It sounds like planting wasp nests around one's house to prevent a wasp invasion. The whole article is confusing actually. I didn't understand it until way after, when you showed fig. 7.

We are sorry for the confusion. We take from this comment that the reviewer finally (fig. 8 – former fig. 7) understood the mechanism but not in the paragraph were it was described. As a consequence we re-wrote the paragraph (but also the introduction paragraph on lowering/increasing f around anticyclonic eddies and the impact on the propagation of NIW). It is also of important to mention that we wrongly interpreted the $f_{eff}$ distribution. The lowering of the planetary vorticity in the core of ACME/AC creates superinertial NIW that propagate downward. In the $f_{eff} > f$ region at the transition zone between the eddy and the surrounding waters  ("ridge region", Halle and Pinkel 2003) NIW energy propagates downward until reflected of dissipated (see e.g. Halle and Pinkel 2003; Fig. 16). This correction also required some

modification in figure 8 (former Fig. 7) – which might be appreciated by this particular reviewer mentioning some concerns with the graphical realisation.

P10L4-12: I am not sure what this paragraph is about. My take on it: does mixing work differently for nutrients than it does for other quantities? But I'm still unsure of the answer.

Our intention was to discuss differences in surface signatures of nutrient upwelling (primary productivity) – is it more at the edge of an eddy or in the centre? The paragraph did not consider differences in mixing of different quantities. However, we re-wrote the paragraph.

P10L22-29: A bit of ray tracing would not add much work, and could greatly improve the credibility of your hypothesis.

We think that ray tracing that go beyond what has been already shown by Sheen et al. (2015) is outside of the scope of this manuscript. Instead we introduced the gradient Richardson number analysis on the individual CTD stations from the ships survey. The Ri analysis does support our conclusion about where the mixing is and why. Moreover, enhanced mixing at the N2 maximum in an ACME was also recently reported for an Arctic eddy (Kawaguchi et al. 2016) and that further support our interpretation of the data. What we actually miss in our observations is microstructure data that would help to quantify the mixing efficiency across an ACME – this data is to be collected in the future.

P11L14-26: my take from this paragraph: there is now an exchange pathway between the mixed layer and the core. Then what about everything you said in the preceding paragraphs? Is there a contradiction or is this a different issue?

The exchange is focussed at the mixed layer base (but only outward) and at the rim or edge of the eddy. There is no evidence from our data that support an exchange of the core (inward from the $N^2$ maxima) with the surrounding water. The term "erosion" should emphasize that the mixing is just towards the outside of the core - "outward" from the N2 maximum" and the core properties are largely unaffected. The term erosion has been used in the past in describing process that operate at the edge of warm core eddies (citation: "note that lateral intrusion and mixing on the sides of the eddy are contributing most to its erosion" Kroll, 1993).

P12L29-P13L15: Same problem as above. I don't find this paragraph very convincing. It is an interesting scenario, but fig. 7 is not substantiated by diffusive fluxes measurement/estimates. If Beal 2007 actually has something to say about it, you might want to use her article more, not cite her in passing. My suggestion   is that this part be moved to the discussion section, with a much more honest depiction of how little you know about why

some properties are exchanged, and why some others aren't, and with a much more measured use of process-based interpretations (at least for the physical processes; I can't judge the chemistry part).

The new Ri analysis supports the mixing scenarios that are suggested by the property distributions. Analyses of microstructure observations that are interpreted as a result of NIW interaction with ACMEs (e.g. shallow ACME: Kawaguchi et al. 2016; deep ACME Sheen et al. 2015) have been published elsewhere. We followed your advice and move this part to the end of the paper.

P11L11-15: I thought I knew what flux was until I read these sentences. What do you mean by flux? Advective flux, diffusive flux? What do you mean by erosion? What does the phrase 'NO3-/oxygen from the eddy core is primarily outward' mean? Why would a flux necessarily transfer stuff from the outside? Are you talking about a mass flux, which in all rigour should be advective? Or a diffusive flux, in which case you may or may not be right depending on the concentration distribution? And what non-dimensional number quantifies the statement 'erosion rather than flux'?

We suspect you mean P13L11—15? The problem with a gradient flux considering an advective/ diffusive balance is that it would EXCHANGE properties – hence the core would be altered in its properties (e.g. Theta/$S_A$). What we actually observe is a remarkable constant T/S slope (and a decrease in oxygen over time – see figure below) but with a narrowing of the T/S range which indicate an erosion of the core above and below the eddy. The observations of a maximum in mixing efficiency at the N2 maximum by Sheen et al. (2015) and Kawaguchi et al. (2016), combined with the minimum in mixing efficiency in the core of the ACME (that is in-line with the NIW propagation pathways as simulated by Sheen et al. 2015 and Halle and Pinkel 2003) support an "erosion" scenario. With "erosion" we mean a transformation of the water at the N2 maximum into surrounding waters, which in turn drives a "shrinking" of the ACME core. We modified the text accordingly and added Figure 4.

P11L27-32 and figure 5b: are this paragraph and figure the only ones that actually lay out your case for an influence of Ekman transport on the ACME? If so, it is a very weak case, not enough to make it to the body of the article in my opinion, and certainly not enough to make it to the abstract, Once again, it could make it to the discussion section, in passing. Thomas 2005 considers a wildly different parameter regime by the way, I don't see how it can help you support your case without more calculations.

We agree and removed the paragraph.

P09L13: Could the low oxygen property have originated from the coast and simply have been

transported all the way to here? I know that you report a decrease from 8 to 3 micromoles/grams over the course of the experiment, but I don't know the error bars on these measurements. And as far as I can tell, you simply say at some point in the text that the signal looks real or something, but that's not quite the quantified statement, especially since so much hinges on it.

A very valid comment. We could show in the past (Karstensen et al. 2015, Fiedler et al. 2016, Schütte et al. 2016) that the low oxygen core did not originate from the coast. For example, direct observation of an Argo float with oxygen sensor that was trapped in a CE over a period of more than 7 month (Karstensen et al. 2015) from the upwelling region into the open North Atlantic showed a constant decrease in oxygen in the eddy core. Also from a number of direct observations of eddies that were surveyed shortly after they detached from the coast and many month later again (Karstensen et al. 2015, Fiedler et al. 2016, Schütte et al. 2016).

P14L23-25: 'The NIW concept (. . .) numerical models': it depends on which models you're talking about. Numerical process studies could resolve these sorts of scales (for a low-res version of what is achievable, see 10.1175/JPO-D-14- 0097.1; I am not an author, and I am not suggesting that you cite it), and could be the most obvious types of studies that could substantiate the viability of your hypotheses. So, I'd like this sentence to be rephrased in order to sound less like 'mission accomplished' and more like 'idealised process studies are needed'

Thank you for the comment. Of course there are models that do resolve the scales and hopefully the processes. We rephrased the sentence accordingly.

**Technical Comments:**

The Text has been modified throughout and it is difficult to identify the commented passages below. However, please be sure we applied the suggested changes BEFORE the text was re-written.

P01L14: extending from about 60 to 200 m depth and. . .?  - done

P01L21: possibly  -done

P02L03: 'has been conducted' => 'were conducted by Chaigneau. . .'  - sentence removed

P02L10-13: you are describing a vertical stacking, or a baroclinic structure. Took me a while to figure out that it wasn't a radial shielding structure. And what do you mean by 'normal'?

Surface-intensified or barotropic? I don't see why one is more normal than the other anyway. I would also talk about ACEs rather than AEs, to be in line with ACMEs. And can't there be CMEs? – We rephrased the sentence and hope it is now clear that describe the stratification and the Mode. In the context of water masses the word "Mode" is often used for nearly homogenous properties such as for subtropical, or subpolar Mode Waters. We are not aware of a publication on "Cyclonic Modewater eddies" but would be happy to add a reference if the reviewer could provide one.

P02L26-29: something odd in that sentence. Perhaps the wrong verb ('explains') is used, or a comma is missing between 'ACME' and 'with', but something is odd. - We rephrased the sentence.

P03L04: 'Mesoscale eddies often have Ro close to 1' => 'Although usually characterised by Ro « 1, mesoscale eddies often feature local values of Ro closer to one'. See my Special Comment 1 though: you might disagree with me. – We rephrased the sentence accordingly.

P03L25-26: 'the modelled . . . eddy core.' If that's the message of the paragraph, it should be placed at the beginning. – We rephrased the whole paragraph.

P03L29: by rim, do you mean top/bottom or lateral rim or both? I would say edge or boundary actually. Rim sounds like lateral boundaries, which is what you might be referring to. – We rephrased the sentence

P04L15: 'and that' => 'which' – We rephrased the whole paragraph.

P04L26: 'but purely opportunistic': huh? I think you can delete anyway, no one is judging. - was deleted

P06L10: SA => $S_A$ -changed

P07L16-17: 'During the last survey. . . 120 m': I actually see two minima, both at 120 m. Do you mean in the vertical again? - The sentence referred to the vertical and we modified the sentence.

P08L06: I don't see how the spiciness section shows the contrasting impact of Theta and $S_A$ on isopycnals. I don't see Θ at all actually, and I don't remember the definition of spice. -Spiciness is constructed to be a variable that is most sensitive to isopycnal thermohaline variations, and least correlated with the density field (Flambert 2002). Because both, Theta and SA, contribute to density it is redundant to analyse them jointly along isopycnals. However, we decided to remove the spiciness discussion and added Theta section instead.

P08L13: 'but separating the eddy surrounding water from. . .' => 'but well separates the eddy core from the surrounding waters'. –we rephrased  the sentence

P08L16: in the stability ratio, what is the z index supposed to mean? Besides, you mix up θ and Θ here and in subsequent lines.  –It is the vertical gradient/contribution. An explanation for z was now added to the text. Thank you for mentioning the problem with θ and Θ, which is related to using the word Equation editor or the symbol set.

P09L09: 'but for the deeper levels more' => 'but more for the deeper levels'?  - Changed accordingly

P09L30: 'downward also': missing word in-between?  - Changed sentence

P09L31: a word on what a typical AE stratification is?   - Sentence was removed (whole section rephrased)

P09L34: 'and that also' => 'which also'   - Changed sentence

P10L4-5: 'Having explained the isolation as..., it is tempting to...'   - Sentence was removed (whole section rephrased)

P10L06: what do you mean, 'concept'? conceptual model?   - Sentence was removed (whole section rephrased)

P11L14: 'Only vertical propagation of internal waves does not generate mixing, but (...)' => 'Vertical propagation of internal waves by itself does not generate mixing. In order to do so, . . .'  - Changed sentence

P11L15: I find it hard to conceive critical layer absorption not followed by KH.   - That is true and we changed the sentence

P11L19-20: 'Here the mean . . . vertical mixing': I don't understand this sentence.  -Changed sentence: "Here the mean flow could gain energy from the NIW current that in turn could lead to energy dissipation because of the shear-instability (Kawaguchi et al. 2016)"

P12L25: Is the double minus in NO3- intentional?  -Typo, Changed

P13L04: what's PON? – It stands for "Particulate Organic Nitrogen"  (which is nitrogen that is part of particles made out of organic substances) – we added the explanation.

Fig. 7: A few of my colleagues (not in this field) and I unanimously agree: this figure looks too

much like a particular piece of anatomy. We all suggest that you change the aspect ratio, make it less symmetric, and/or replace the blue and yellow lines by different lines. Once seen, it can't be unseen. Besides that, I thought oxygen was not transported in an out of the eddy (P14L15), so what's up with the yellow lines? I'd also like to see arrowheads on the blue and yellow lines, even if bi-directional (I don't think they would be). Finally, I'd like to see the huge converging arrows towards the centre of the eddy removed. I get it that some stuff is retained inside the eddy, but let's not forget that in a vortex, geostrophic or not, velocities are mostly azimuthal. I understand that this is meant to reinforce your point, but in the end, it is misleading. Or make them squiggly, which would evoke diffusion. - We changed the figure completely and hope it does not any longer displease peoples eyes. The low oxygen (high nitrate) waters are eroded from the core and introduced into the mixed layer. Here, low oxygen is brought back towards saturation by air/sea gas exchange – but this is not true for nitrate. The nitrate is taken up by phytoplankton and is reintroduced in the core via gravitational sinking of Particulate Organic Nitrogen (PON) into the core (new Fig 6 a). Back in the core, one certain nitrate molecule can be used multiple times for respiration of oxygen. This alters the AOU:NO3 ratio high.

Another important consideration is that I needed to read carefully four manuscripts within the special issue to deeply understand the results and the discussion, the manuscript (ms) is full of typos or miss- references to the figures. It seems that the authors did not check the ms coherence before submitting, this is a very bad point for their reputation. Considering the amount of coauthors an effort should have been done to ease the reading of the ms and make it a stand- alone work.

Despite this I think the ms merits to be published after some improvements both in content and layout. I understand that it is somehow difficult to organize the wealth amount of data recorded by the different surveys and observing platforms deployed to characterize this intriguing new dead zones in the ETNA. In addition this paper is mostly about physical oceanography, and I am a chemical oceanographer, maybe the ms needs a third opinion.

We thank the reviewer for encouraging but also critical words. Indeed it is a problem to publish papers that address the interaction of physical and biogeochemical processes. To our opinion the Guest editor did a great job in selecting reviewers that, although addressing primarily points related to their own discipline (Reviewer 1 being a physical oceanographer, Reviewer 2 being a chemical Oceanographer), asked apparently "simple questions" on the others discipline which are often the trickiest to answer. As Reviewer 2 will see, we have revised the text and reordered the content over many parts. We reworked the figure and added one new figure that nicely shows the sinking of particles into the eddy core. We hope that the new version is now focussing to the point and thus is easier to read. We tried to simplify the physical and biogeochemical parts and reworked the summary figure (now Fig. 8). However, certain parts might be difficult to fully understand by one or the other disciplinary reader.

Indeed the many typos in the submitted version go fully to the account of the lead author. In fact, the Guest editor had kindly provided a proofread version that could have been used for publication – but unfortunately the file was "overlooked" by the lead author in the submission process.

A fundamental issue is the prime hypothesis of this ms which is finally resolved in Fig.7, the authors propose a physical mechanism to explain the isolation of the eddy core but also another one (near inertial waves, NIW, breaking) to explain the flux of nutrients to the upper mixed layer. As the authors say in the text the evidences to support the physical mechanisms

suffer from "not having concurrent hydrography and currents data and limited options for estimating balances" (P14, L3-4). On the biogeochemical side, the authors only support their "nitrogen cycling" hypothesis with nitrate and oxygen data from the glider surveys, but other measurements are available from typical CTD casts as described in Fiedler et al. (BGD 2016).

We added now a more detailed analysis of the potential mixing sites from a more „in depth" analysis of the critical Richardson number. Moreover, a number of references are added that support the erosion concept (Halle and Pinkel 2003, Kawaguchi et al. 2016, Kroll 1993).

We added one figure that nicely shows the high particle load that "rains" into the eddy core (Fig. 6) and which further supports the idea that particle sinking and remineralization is one key process in creating the low oxygen core.

We also now added NO3/AOU data from the ship cruise that exactly show the same alteration of the AOU:NO3 ratio in the core.

**SPECIFIC COMMENTS**

Introduction
Although the intro is rather long, just the last three lines contain some references to the other ms related to the studied Anticyclonic Mode Water Eddies (ACME) within the same project and using the same observing platforms. I think a comprehensive summary of the different genomic, biological and biogeochemical aspects of the ACMEs should be given, also highlighting the contribution of the current ms.

We re-wrote and restructured the introduction. In particular the eddy detection part was removed (irrelevant for the present study). Details about the different genomic, biological and biogeochemical aspects of the particular ACME that we investigate here will be given in an overview article for the special issue (In preparation).

2. Data and methods
2.1. Glider survey
Maybe a word or reference about the interpolation method for the glider data would be interesting.
A linear interpolation was applied (now added to the text)

2.2 Glider sensor calibration
Page5, line 16. I would like to see some number about oxygen precision and accuracy, as done for nitrate (P6, L7-8). Although more details about this are surely given in Hahn et al 2014, please consider my demand.

The comparison between calibrated (by titration of oxygen samples) Clarke sensor on the CTD and the calibrated optode data suggests an overall (full oxygen range) RMS error of 3 μmol

kg−1. However, for the chemically forced 0 µmol kg−1 oxygen an RMS error of 1 µmol kg−1 is expected. We added the information to the text.

2.3. Ship survey
I do not understand why not using the biogeochemical data gathered during M105, at least NO3, PO4, O2, particulate and dissolved organic matter, to sustain your biogeochemical interpretation of the results. More comments about this issue will be given in the corresponding section of the ms.
-As outlined under the specific points below, we primarily reference to the published figures in the accompanying articles. However, we added the AOU/NO3 data from the profiles taken un the eddy (ISL_000314).

3. Results and Discussion
3.1 Vertical Eddy Structure "Biogeosciences" is not "Journal of Physical Oceanography" so my excuses for not understanding all the difficult terms in this section. As the aim of the ms is explaining the "fluxes of nitrate" into the mixed layer supporting the high primary production in the ACME, my opinion is that an effort should be done to make the ms more readable for the ocean biogeochemical community.
- The aim of the manuscript is to better understand mixing and isolation AND to reflect this back to the nitrogen cycling. We hope that the extensive rewriting of the manuscript has fixed this issue (also by summarizing the results in Figure 8 in a more transparent way).

P9L5-9. I checked (I read) Fiedler et al 2016 and I did not find any explanation about the translational velocity of the ACME, I found this information in Karstensen et al (BG 2015).
In section "3.1. Eddy Characteristics" Fielder et al. discuss the translation velocity. However, numbers are given in Karstensen et al. (2015) and Schütte et al. (2016). We changed accordingly.

3.2 Eddy core isolation and vertical fluxes. Please check the figure references in this section, it is a mess!! It was very hard to follow the result description and the final message to be conveyed.
The section has been completely rewritten and figures are re-done. We hope it is now easy to understand.

P9-L13 no reference to limnic systems is given in Karstensen et al (2015).  -In the abstract of Karstensen et al. (2015) it says: "…create the "dead zone" inside the eddies, so far only reported for coastal areas or lakes." – with "lakes" we anticipated limnic systems, but maybe that is incorrect?

P9-L19: the nonlinearity parameter is not defined or commented previously in this work but in Karstensen et al (2015). Please explain why alpha is important for the coherence of the eddy but it does not matter to explain isolation. -Eddies might be separated into linear waves (also called "Rossby Waves") or in isolated, coherent structures. The non-linearity parameter is a measure to judge if the feature is a linear wave (translation speed and rotation speed are similar) or a coherent eddy (rotation speed much higher than translation) with a different dynamical regime. We refer in the text to the isolation of the core against lateral or vertical mixing – maybe

shielding for mixing is a better formulation? The paragraph has been rewritten and a reference added (Chelton et al. 2011).

P10-L2-3. Weird phrase. – Indeed, but the paragraph has been completely rewritten and the sentence is removed.

P11. A mess with the figure references. Please just for the biogeochemist summarize where would NIW brake and induce mixing / fluxes in the eddy structure.
- We rephrased the sentences and hope that we provide with (new) Figure 8 a good overview about were exactly mixing occurs. We structured in three regions (I, II, III).

P11-L8-9. "no concurrent velocity and stratification section data exists" I do not understand, you have velocity and CTD casts from the ship so at least you have 8 stations. –We now exclusively use the CTD and ADCP shear data for estimating the gradient Richardson number (P11L8) and added information into Figure 4 (d, h).

3.3 Nutrient budget.
This section should be entitled "nitrate budget"... but not even so... as no budget is estimated, a better title would be "nitrate cycling" .
- This is true and we changed that accordingly

My main concern about this section the rejection of using other biogeochemical data from the ship surveys within the ACMEs. For example why not using the M105 $NO_3$ and AOU data in Fig 6c?, they crossed the eddy center as showed in Fig 2b.
– We added the data from ISL314 to the plot.

An evidence of denitrification would be a differential $NO_3:PO_4$ ratio.
- The $NO_3:PO_4$ ratio has been presented in Löscher et al. (2015) in Figure 3c and in Grundel et al. (in revision, Scientific Reports). It was determined in these papers that the nitrogen loss is in nanomol/kg range – while we talk about 4-6 micromol/kg.

After reading several times this section, the main question is how are the nutrients injected into the mixed layer to support primary production?. However no profile of chlorophyll is given (I found some info about this in Loscher et al. BG 2015) , I wonder if the gliders have at least a backscattering or fluorometer sensor.

- We added a new figure on turbidity that shows the particle sinking (turbidity) across the mixed layer base and deep into the low oxygen core. The fluorescence data we found not very instructive (see below). During the first occupation the high chlorophyll was more homogenous distributed across the mixed layer (but the quenching effect is therefore very pronounced). During the second occupation (IFM13) a clear subsurface Chl maximum relative well aligned with the mixed layer base is found (but the maximum is at about 10m shallower depth). We could add these plots to the Turbidity figure if the reviewer insists.

[Figure]

The biogeochemical info in Fiedler et al BGD 2016 in the shelf, CVOO and the eddies may help to explain the high primary production (PP), if eddies are formed in the shelf, they contain nutrients that are used and converted into organic matter (particulate and dissolved) that sinks and is remineralized in the eddy creating the O2 minimum. Is it enough the initial NO3 in the shelf to sustain PP in the eddy when it moves into the ETNA?. Does it really need an extra NO3 input?

- It is not a problem of the initial nutrient content in the eddy core but the processes that transport nutrients into the euphotic zone of the eddy. The process we propose alters fundamentally the NO3:AOU ratio because it is based on recycling of nutrients (but not of oxygen), which in turn is the results of the specifics of upwelling in the eddy.
We would not call that an "extra NO3" but a recycling that alters the AOU/NO3 ratio. Figure 4 in Fiedler et al. (2016) shows the profiles from the cruises as well as from the shelf. It can be seen that the shelf water are lower in NO3 (and higher in O2) than the waters in the eddy core. Moreover, in Karstensen et al. (2015) we could show that the low oxygen is generated en-route and not in the eddy at the time when it detaches from the coast.

It is very hard to understand a decoupled O2 and NO3 cycle if denitrification is not important. Please check the NO3:PO4 ratio. An anomalous O2:NO3 ratio could be related to the stochiometry of the organic matter remineralized both particulate and dissolved, please check the available data.

- As mentioned earlier, Löscher et al. (2015) (and in Grundle et al. in revision, Scientific Reports) determined the nitrogen loss by denitrification to be in the nanomolar range but the nitrate deficit is in the micromolar range.

We explain the decoupling from the specific mixing pattern of the eddy (erosion of the core) and the differences in what happens to oxygen and nitrate when reintroduced into the mixed layer (euphotic zone) of the eddy. Once in the mixed layer the pathways of the high NO3 and the low O2 water are different. The low O2 water will lower the oxygen content in the mixed layer (being now undersaturated in oxygen) but which is recharged by air/sea gas exchange. In contrast, the high NO3 water of the core will provide new nutrients to the mixed layer and in turn stimulate productivity. The nitrogen is incorporated into particles (as PON) in the productivity processes, which in turn sink out of the mixed layer and back into the core. Through this pathway, some of the upwelled nitrogen is re-entering the eddy core and is ready for driving new respiration via remineralization. Essentially the reintroduction of NO3 is a gravitational process and O2 does not participate in it.

4. Summary and conclusions
I suppose it would need to be rewritten depending on the results from section 3.3.
-We have rewritten the section 4 and hope to made the points now more clear.

I hope to have been helpful.
-Definitely – Thank you!

[revised manuscript text omitted]

Johannes 16/2/2017 16:49
[4] verschoben (Einfügung) ... [63]

Johannes 16/2/2017 16:49
[5] verschoben (Einfügung) ... [65]

Johannes 16/2/2017 16:49
[2] verschoben (Einfügung) ... [66]

Johannes 16/2/2017 16:49

Johannes 16/2/2017 16:49

Johannes 16/2/2017 16:49

Johannes 16/2/2017 16:49

Johannes 16/2/2017 16:49

Johannes 16/2/2017 16:49

Johannes 16/2/2017 16:49

Johannes 16/2/2017 16:49

Johannes 16/2/2017 16:49

Johannes 16/2/2017 16:49

Johannes 16/2/2017 16:49

Johannes 16/2/2017 16:49

Johannes 16/2/2017 16:49

Johannes 16/2/2017 16:49

In anticyclonic rotating eddies the downward propagation of wave energy in the $f_{eff} < f$ region has been observed and modelled (Kunze 1985, Gregg et al. 1986, Lee and Niiler 1998, Koszalka et al. 2010, Joyce et al. 2013, Alford et al. 2016). Lee and Niiler (1998) simulated the NIW interaction with eddies (ACE, CE, ACME) and found vertical propagation of the NIW energy, the "inertial chimney". In the case of an ACME with a low squared buoyancy frequency ($N^2$) layer they report on NIW energy accumulating below the eddy core and not inside as seen for ACE. This change in energy accumulation was attributed to the vertical stratification of the ACME, in particular the two $N^2$ maxima that shield the low stratified eddy core. Kunze et al. (1995) analysed NIW energy propagation in an ACE. Within a critical layer at the inner sides of the ACE and where the $f_{eff}$ increases $\geq 1$, the vertical propagation of NIWs is hampered and energy accumulates, the bulk is being released by turbulent mixing.

The vertical shear of the horizontal velocity that is generated by NIWs can eventually force overturning e.g. which approaching a critical layer (Kunze et al. 1995). The tendency of a stratified water column to become unstable through velocity shear $S = \sqrt{\left(\frac{\partial u}{\partial z}\right)^2 + \left(\frac{\partial v}{\partial z}\right)^2}$ can be estimated from the gradient Richardson number $Ri = N^2/S^2$. A $Ri < 1/4$ has been found a necessary condition for the shear to overcome the stratification and to generate overturning. However, Whitt et al. (submitted) measured enhanced dissipation was with shear probes along the Gulf Stream front in several regions where NIW shear produced $Ri < 1$.

Recent observational studies using microstructure shear probe data report enhanced mixing in a narrow depth range of a local, vertical $N^2$ maximum, above and below the low stratified ACME core (Sheen et al. 2015, Kawaguchi et al. 2016). By applying a internal wave ray trace model to the $N^2$ stability profile and vertical shear profile from outside and from inside of an ACME, Sheen et al. (2015) could show that only a very limited range of incident angles of internal waves could propagate into the eddy core. Most NIWs encounter a critical layer above and below the eddy, the regions where they observed enhanced mixing. Halle and Pinkel (2003) analysed NIW interaction with eddies (owning a ACME vertical structure) in the Arctic and explained the low internal wave activity in the core as the result of an increase (order of magnitude) in wave group speed caused by low $N^2$ but which in turn lowering of wave energy density. Krahmann et al. (2008) reported observations of enhanced NIW energy in the vicinity of a Meddie. For Meddies signatures of thermohaline layering at the eddy periphery have often been observed and occurrence of critical layers identified that support the energy transfer from the mesoscale to the submesoscale (Hua et al. 2013).

In this paper we investigate the hydrography, currents, and biogeochemical characteristic of a low oxygen ACME and its temporal evolution. High-resolution underwater glider and ship data allow us to describe the eddy structure at submesoscale resolution. Characteristics of a low oxygen ACME found in the eastern tropical North Atlantic are provided. The paper is part of a series of publications that report

Johannes 16/2/2017 16:49
Johannes 16/2/2017 16:49
Johannes 16/2/2017 16:49
Johannes 16/2/2017 16:49
Johannes 16/2/2017 16:49
Johannes 16/2/2017 16:49
Johannes 16/2/2017 16:49
Johannes 16/2/2017 16:49
Johannes 16/2/2017 16:49
Johannes 16/2/2017 16:49
Johannes 16/2/2017 16:49
Johannes 16/2/2017 16:49
Johannes 16/2/2017 16:49
Johannes 16/2/2017 16:49
Johannes 16/2/2017 16:49
Johannes 16/2/2017 16:49
Johannes 16/2/2017 16:49

[revised manuscript text omitted]

Johannes 16/2/2017 16:49
Johannes 16/2/2017 16:49
Johannes 16/2/2017 16:49
Johannes 16/2/2017 16:49
Johannes 16/2/2017 16:49
Johannes 16/2/2017 16:49
Johannes 16/2/2017 16:49
Johannes 16/2/2017 16:49
Johannes 16/2/2017 16:49
Johannes 16/2/2017 16:49
Johannes 16/2/2017 16:49
Johannes 16/2/2017 16:49
Johannes 16/2/2017 16:49
Johannes 16/2/2017 16:49
Johannes 16/2/2017 16:49

*0dbar) and mapped to a linear section in latitude, longitude. In b) the yellow dots indicate positions of local $N^2$ maxima in the CTD profiles.*

During the first survey (IFM12), lowest oxygen concentrations of about 8 μmol kg$^{-1}$ were observed in two vertically separated cores at about 80 m and 120 m depth, while in between the two cores, oxygen concentrations increased to about 15 μmol kg$^{-1}$. About 6 weeks later, during the M105 ship survey, lowest concentrations of about 5 μmol kg$^{-1}$ were observed, centred at about 100 m depth and without a clear double minimum anymore, based on six CTD stations. During the last glider survey (IFM13), another three weeks after the ship survey, the minimum concentrations were < 3 μmol kg$^{-1}$ and showed in the vertical a single minimum centred at about 120 m. The intensification of the minimum (by about 5 μmol kg$^{-1}$ in 2 months) is assumed here to be a result of continues respiration without balancing lateral/vertical mixing oxygen supply. It is important to note that during the glider survey the eddy performed about one full rotation and hence we expect less impact of the spatial variability in our sampling of the core. Underneath the eddy core, and best seen in the 40 μmol kg$^{-1}$ oxygen contour below 350 m at about 0 km (centre), an increase in oxygen over time is found indicating supply of oxygen from surrounding waters. Comparing the two glider surveys (Fig. 2) a broadening of the gradient zone at the upper boundary of the core is observed. Overall the upper boundary of the core during the first survey aligned tightly with the mixed layer base giving the core the shape of a plan convex lens, while the lens developed into a biconvex shape before the second glider survey (also seen in the ship survey Fig. 2b).

The SLA data analysis for the eddy (see Schütte et al. 2016b for details) suggests a formation in the Mauritanian upwelling region (Fig. 1). The composite of the outermost ("last") closed geostrophic contour of the eddy (Fig. 1, right), revealed a diameter of about 60 km, which is in accordance with the dimensions of the vertical structures observed from the glider and the ship (Fig. 2 and 3). The eddy core is composed of a fresh and cold (Fig. 3a, b; Fig 4a) water mass that matches the characteristics of South Atlantic Central Water (SACW; Fiedler et al. 2016) and is a typical composition for low oxygen eddies in the eastern tropical North Atlantic (Karstensen et al. 2015, Schütte et al. 2016a, 2016b). The properties confirm that the ACME was formed in the coastal area off Mauritania, as suggested by the SLA analysis (Schütte et al. 2016b; Fig. 1, left). Layering of properties, as seen in oxygen (Fig. 3), is also observed in $S_A$ and $\Theta$ (Fig. 4) underneath the eddy core. In the depth range between 160 to 250 m the layers are aligned with density contours and suggest isopycnal transport processes, while below that depth range, and at the edges of the eddy core the thermohaline intrusions cross density surfaces.

The low oxygen core of the ACME is well separated from the surrounding water through maxima in $N^2$ (Fig. 3c). The most stable conditions ($N^2$ about 3 to 5*10$^{-4}$ s$^{-2}$; compare Fig. 4a) are found along the upper boundary of the core and aligned with the mixed layer base (changed from 50 to 70 m between

IFM12 and IFM13, respectively). Θ ($S_A$) differences across the mixed layer base were large, about 5 to 6 K (0.7 – 1.0 gr kg$^{-1}$), but over time the mixed layer base widened from 8 m (glider IFM12 survey) to 16 m (M105 ship) and to 40 m (glider IFM13 survey) (Fig. 4a).

[Figure]

Johannes 16/2/2017 16:49

Johannes 16/2/2017 16:49

Johannes 16/2/2017 16:49

Johannes 16/2/2017 16:49

Johannes 16/2/2017 16:49

Johannes 16/2/2017 16:49

Johannes 16/2/2017 16:49

Johannes 16/2/2017 16:49

Johannes 16/2/2017 16:49

Johannes 16/2/2017 16:49

Johannes 16/2/2017 16:49

Johannes 16/2/2017 16:49

Johannes 16/2/2017 16:49

Johannes 16/2/2017 16:49

Johannes 16/2/2017 16:49

[revised manuscript text omitted]

Johannes 16/2/2017 16:49
Johannes 16/2/2017 16:49
Johannes 16/2/2017 16:49
Johannes 16/2/2017 16:49
Johannes 16/2/2017 16:49

revision, Scientific Reports), and not necessarily for overall $NO_3^-$ losses which are measured in the micromolar range. Grundle et al. (in revision, Scientific Reports) showed by relating nitrogen and phosphorous cycling, that in the core of the ACME the $NO_3^-$ losses were not detectable at the micromolar range. Thus, while denitrification may have played a minor role in causing the higher than

5  expected $AOU:NO_3^-$ ratio which we have calculated, it is unlikely that it contributed largely to the loss of 5% of all $NO_3^-$ from the eddy as estimated based on the observed $AOU:NO_3^-$ ratios.

Alternatively, but perhaps not exclusively, the $NO_3^-$ recycling within the ACME could be the reason for the $NO_3^-$ deficit. A high $AOU:NO_3^-$ ratio could be explained through a decoupling of $NO_3^-$ and oxygen recycling pathways in the eddy and details about the vertical transport pathways of nutrients (erosion of

10  core). Based on the investigation of the possible vertical mixing/transport of nutrients (here $NO_3^-$) the erosion of the eddy core plays a key role. In this scenario $NO_3^-$ molecules are used more than one time for the remineralization/respiration process and therefore the AOU increase without a balanced, in a Redfieldian sense, $NO_3^-$ remineralization. Such a decoupling can be conceptualized as follows (Fig. 8, left): consider an upward flux of dissolved $NO_3^-$ and oxygen in a given ratio with an amount of water

15  that originates from the low oxygen core. The upward flux partitions when reaching the mixed layer, one part disperses in the open waters outside of the eddy, the other part is keep in the eddy by retention (D'Ovidio et al. 2013). The upwelled $NO_3^-$ is utilized by autotrophs for primary production and thereby incorporated into particles as Particulate Organic Nitrogen (PON) while the corresponding oxygen production is re-ventilated by air/sea oxygen flux. The PON sinks out of the mixed layer/euphotic zone

20  and into the core of the eddy were remineralization of organic matter releases quickly some of $NO_3^-$ originating from the core waters. In contrast, the upwelling of oxygen-deficient waters will drive an oxygen flux from the atmosphere into the ocean in order to reach chemical equilibrium. But because the stoichiometric equivalent of oxygen is exchanged with the atmosphere and therefore not transported back into the core by gravitational settling of particles, as it is the case for nitrate (via PON), the

25  respiration associated with the remineralization of the recycled nitrate will results in a net increase in AOU.

**4 Summary and Conclusion**

Here we present a first analysis of the temporal evolution of a low oxygen ACME in the eastern tropical North Atlantic from high-resolution multidisciplinary glider and ship survey data. The low oxygen eddy

30  has a diameter of about 70 to 80 km and a maximum swirl velocity of 0.4 m s$^{-1}$ close to the mixed layer base and can be considered typical for the region (Schütte et al. 2016a, 2016b). The eddy originated from the Mauritanian upwelling region (Schütte et al. 2016b; Fiedler et al. 2016) and had a distinct anomalously fresh and cold water mass in its low oxygen core. The core was located immediately below the mixed layer base (about 70 to 80 m) down to a depth of 200 to 250 m in its centre. The core

Johannes 16/2/2017 16:49

Johannes 16/2/2017 16:49

Johannes 16/2/2017 16:49

Johannes 16/2/2017 16:49

Johannes 16/2/2017 16:49

Johannes 16/2/2017 16:49

Johannes 16/2/2017 16:49

Johannes 16/2/2017 16:49

Johannes 16/2/2017 16:49

Johannes 16/2/2017 16:49

Johannes 16/2/2017 16:49

Johannes 16/2/2017 16:49

Johannes 16/2/2017 16:49

Johannes 16/2/2017 16:49

Johannes 16/2/2017 16:49

Johannes 16/2/2017 16:49

Johannes 16/2/2017 16:49
... [418]

Johannes 16/2/2017 16:49

[revised manuscript text omitted]

---

## Editor Decision (ED1)

November 23, 2016

Dear Dr Karstensen and co-authors,

After carefully reading the comments from both referees, as well as your detailed responses to those comments, I have come to the conclusion that major changes to your manuscript are necessary before it can be accepted for publication in Biogeosciences. The two referees from the first round of reviews have provided very constructive comments that I think will help clarify the main points of the paper. I will seek their opinions on the revised paper and your responses to their comments, provided their schedules allow them to be reassigned as referees.

In addition to the reviewers' comments, I would appreciate if you could consider the two points below.

- Given that the eddy core described by Sheen et al. (2015) is located at a much greater depth (about 2000 m) than the 125 m deep core of the anticyclonic mode-water eddy (ACME) of your study, I believe that ray tracing specifically done for your study's ACME would help better determine the critical layer depth(s). See Referee # 1, Specific comment # 5;
- Is vertical mixing really taking place mostly on the lateral edges of the eddy, as shown on your Fig. 7, rather than near the top of the eddy? Figure 4c of Sheen et al. (2015) shows enhanced mixing at the top of the eddy, at about the same depth (1500 m in their case) as where their critical layer scenario is located (their Fig. 5c).

Best regards,

Denis Gilbert, Guest editor